# Policy assessments for the carbon emission flows and sustainability of Bitcoin blockchain operation in China

Shangrong Jiang [1,7], Yuze Li [1,2,7], Quanying Lu[2], Yongmiao Hong[2,3], Dabo Guan[4 ✉], Yu Xiong[5] & Shouyang Wang [1,2,6 ✉]

The growing energy consumption and associated carbon emission of Bitcoin mining could potentially undermine global sustainable efforts. By investigating carbon emission flows of Bitcoin blockchain operation in China with a simulation-based Bitcoin blockchain carbon emission model, we find that without any policy interventions, the annual energy consumption of the Bitcoin blockchain in China is expected to peak in 2024 at 296.59 Twh and generate 130.50 million metric tons of carbon emission correspondingly. Internationally, this emission output would exceed the total annualized greenhouse gas emission output of the Czech Republic and Qatar. Domestically, it ranks in the top 10 among 182 cities and 42 industrial sectors in China. In this work, we show that moving away from the current punitive carbon tax policy to a site regulation policy which induces changes in the energy consumption structure of the mining activities is more effective in limiting carbon emission of Bitcoin blockchain operation.

[1] School of Economics and Management, University of Chinese Academy of Sciences, Beijing, China. [2] Academy of Mathematics and Systems Science, Chinese Academy of Sciences, Beijing, China. [3] Department of Economics and Department of Statistical Science, Cornell University, Ithaca, NY, USA. [4] Department of Earth System Science, Tsinghua University, Beijing, China. [5] Surrey Business School, University of Surrey, Guildford, UK. [6] Center for Forecasting Science, Chinese Academy of Sciences, Beijing, China. [7]These authors contributed equally: Shangrong Jiang, Yuze Li.
✉email: guandabo@tsinghua.edu.cn; sywang@amss.ac.cn

As Bitcoin attracted considerable amount of attention in recent years, its underlying core mechanism, namely blockchain technology, has also quickly gained popularity. Due to its key characteristics such as decentralization, auditability, and anonymity, blockchain is widely regarded as one of the most promising and attractive technologies for a variety of industries, such as supply chain finance, production operations management, logistics management, and the Internet of Things (IoT)[1–3]. Despite its promises and attractiveness, its first application in the actual operation of the Bitcoin network indicates that there exists a non-negligible energy and carbon emission drawback with the current consensus algorithm. Therefore, there is an urgent need to address this issue. In this paper, we quantify the current and future carbon emission patterns of Bitcoin blockchain operation in China under different carbon policies. In recent years, the system dynamics (SD) based model is widely introduced for carbon emission flow estimation of a specific area or industry[4,5]. In comparison to its counterparts, SD modeling has two main advantages in carbon emission flow assessment: first, by combining the feedback loops of stock and flow parameters, SD is able to capture and reproduce the endogenous dynamics of complex system elements, which enables the simulation and estimation of specific industry operations[6–8]. In addition, since the SD-based model is focused on disequilibrium dynamics of the complex system[9,10], intended policies can be adjusted for scenario policy effectiveness evaluation. Consequently, based on system dynamics modeling, we develop the Bitcoin blockchain carbon emission model (BBCE) to assess the carbon emission flows of the Bitcoin network operation in China under different scenarios.

This paper uses the theory of carbon footprint to create a theoretical model for Bitcoin blockchain carbon emission assessment and policy evaluation[11,12]. First, we establish the system boundary and feedback loops for the Bitcoin blockchain carbon emission system, which serve as the theoretical framework to investigate the carbon emission mechanism of the Bitcoin blockchain. The BBCE model consists of three interacting subsystems: Bitcoin blockchain mining and transaction subsystem, Bitcoin blockchain energy consumption subsystem, and Bitcoin blockchain carbon emission subsystem. Specifically, transactions packaged in the block are confirmed when the block is formally broadcasted to the Bitcoin blockchain. To increase the probability of mining a new block and getting rewarded, mining hardware will be updated continuously and invested by network participants for a higher hash rate, which would cause the overall hash rate of the whole network to rise. The network mining power is determined by two factors: first, the network hash rate (hashes computed per second) positively accounts for the mining power increase in the Bitcoin blockchain when high hash rate miners are mining; second, power usage efficiency (PUE) is introduced to illustrate the energy consumption efficiency of Bitcoin blockchain as suggested by Stoll[13]. The network energy cost of the Bitcoin mining process is determined by the network energy consumption and average electricity price, which further influences the dynamic behavior of Bitcoin miners. The BBCE model collects the carbon footprint of Bitcoin miners in both coal-based energy and hydro-based energy regions to formulate the overall carbon emission flows of the whole Bitcoin industry in China. The level variable GDP consists of Bitcoin miner's profit rate and total cost, which reflects the accumulated productivity of the Bitcoin blockchain. It also serves as an auxiliary factor to generate the carbon emission per GDP in our model, which provides guidance for policy makers in implementing the punitive carbon taxation on the Bitcoin mining industry. Bitcoin blockchain reward halving occurs every four years, which means that the reward of broadcasting a new block in Bitcoin blockchain will be zero in 2140. As a result, the Bitcoin market price increases periodically due to the halving mechanism of Bitcoin blockchain. Finally, by combining both carbon cost and energy cost, the total cost of the Bitcoin mining process provides a negative feedback for miner's profit rate and their investment strategies. Miners will gradually stop mining in China or relocate to elsewhere when the mining profit turns negative in our BBCE simulation. The comprehensive theoretical relationship of BBCE parameters is demonstrated in Supplementary Fig. 1.

We find that the annualized energy consumption of the Bitcoin industry in China will peak in 2024 at 296.59 Twh based on the Benchmark simulation of BBCE modeling. This exceeds the total energy consumption level of Italy and Saudi Arabia and ranks 12th among all countries in 2016. Correspondingly, the carbon emission flows of the Bitcoin operation would peak at 130.50 million metric tons per year in 2024. Internationally, this emission output surpasses the total greenhouse gas emission output of the Czech Republic and Qatar in 2016 reported by cia.gov under the Benchmark scenario without any policy intervention. Domestically, the emission output of the Bitcoin mining industry would rank in the top 10 among 182 prefecture-level cities and 42 major industrial sectors in China, accounting for approximately 5.41% of the emissions of the electricity generation in China according to the China Emission Accounts & Datasets (www.ceads.net). In addition, the maximized carbon emission per GDP of the Bitcoin industry would reach 10.77 kg/USD based on BBCE modeling. Through scenario analysis, we find that some commonly implemented carbon emission policies, such as carbon taxation, are relatively ineffective for the Bitcoin industry. On the contrary, site regulation policies for Bitcoin miners which induce changes in the energy consumption structure of the mining activities are able to provide effective negative feedbacks for the carbon emission of Bitcoin blockchain operation.

## Results

**The energy and carbon emission problem of Bitcoin mining in China.** Although the Proof-of-Work (PoW) consensus algorithm has enabled Bitcoin blockchain to operate in a relatively stable manner, several unexpected behaviors of the Bitcoin blockchain have been detected: first, the attractive financial incentive of Bitcoin mining has caused an arms race in dedicated mining hardware[14]. The mining hardware has evolved through several generations. Initially, miners used the basic Central Processing Unit (CPU) on general-purpose computers. Then, a shift was made to the Graphic Processing Unit (GPU) that offered more power and higher hash rates than the CPU. Finally, the Application-Specific Integrated Circuits (ASICs) that are optimized to perform hashing calculations were introduced. Nevertheless, the rapid hardware development and fierce competition have significantly increased the capital expenditure for Bitcoin mining[15]; second, the Bitcoin mining activity and the constant-running mining hardware has led to large energy consumption volume. Previous literature has estimated that the Bitcoin blockchain could consume as much energy per year as a small to medium-sized country such as Denmark, Ireland, or Bangladesh[16]; finally, the large energy consumption of the Bitcoin blockchain has created considerable carbon emissions (see Supplementary Fig. 2 for details). It is estimated that between the period of January 1st, 2016 and June 30th, 2018, up to 13 million metric tons of $CO_2$ emissions can be attributed to the Bitcoin blockchain[17]. Although the estimate ranges vary considerably, they have indicated that energy consumption of network and its corresponding environmental impacts have become a non-negligible issue.

The growing energy consumption and the environmental impacts of the Bitcoin blockchain have posed problems for many

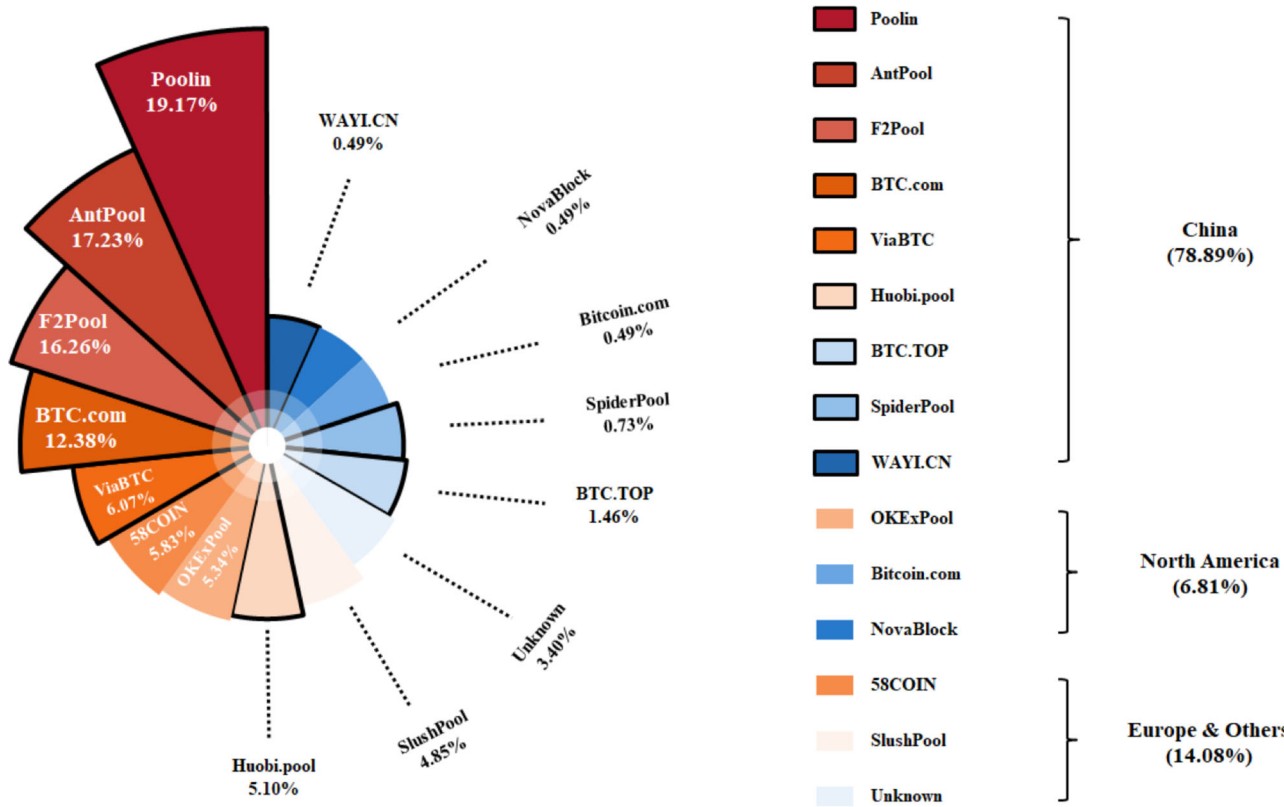

**Fig. 1 Mining pool distributions of Bitcoin blockchain.** As of April 2020, China accounts for more than 75% of Bitcoin blockchain operation around the world. Some rural areas in China are considered as the ideal destination for Bitcoin mining mainly due to the cheaper electricity price and large undeveloped land for pool construction. The mining pool statistics is obtained from https://btc.com/stats.

| Scenarios | Measures | Market access (%) | Miner site selection (%) | Carbon tax |
|---|---|---|---|---|
| Benchmark (BM) | Baseline policy intervention | 100 | 40 | 2 |
| Market access (MA) | Raise the market access standards for Bitcoin miner efficiency | 50 | 40 | 2 |
| Site regulation (SR) | Strict regulation on Bitcoin industry in the coal-based energy region | 100 | 20 | 2 |
| Carbon tax (CT) | Extra punitive carbon tax on Bitcoin mining | 100 | 40 | 4 |

**Table 1 Scenario parameter settings.**

Note: Exogenous auxiliary parameters are introduced to assess the carbon emission flows under different Bitcoin policy measures. In terms of variable settings, three main parameters are chosen as the scenario factors in the proposed BBCE model, including market access (MA), miner site regulation (SR), and carbon tax (CT).

countries, especially for China. Due to the proximity to manufacturers of specialized hardware and access to cheap electricity, majority of the mining process has been conducted in China as miners in the country account for more than 75% of the Bitcoin network's hashing power, as shown in Fig. 1. As one of the largest energy consuming countries on the planet, China is a key signatory of the Paris Agreement[18–20]. However, without appropriate interventions and feasible policies, the intensive Bitcoin blockchain operation in China can quickly grow as a threat that could potentially undermine the emission reduction effort taken place in the country[10].

Suggested by the previous work[21] and the subsystems of our proposed BBCE model, we consider three main Bitcoin policies conducted at different stages of the Bitcoin mining industry, which then formulates the four scenario assessments for Bitcoin blockchain carbon emission flows (in Table 1). In detail, Benchmark (BM) scenario is a baseline and current scenario of each policy factor, which suggests that the Bitcoin industry continues to operate under minimal policy intervention. In the

Benchmark scenario, market access is assumed to be 100%, which indicates that profitable Bitcoin miners of all efficiencies are allowed to operate in China. As suggested by the actual regional statistics of Bitcoin miners, we assume 40% of miners are located in the coal-based area in the Benchmark scenario. Moreover, the punitive carbon tax will be imposed if the carbon emission per GDP of the Bitcoin industry is greater than 2. In the other three scenarios, policies on different Bitcoin mining procedures are adjusted due to energy saving and emission reduction concerns. Specifically, in the Bitcoin mining and transaction subsystem, market access standard for efficiency is doubled, i.e., profitable miners with low efficiency are forbidden to enter the Chinese Bitcoin market in the market access (MA) scenario, and policy makers are forced to maintain the network stability of Bitcoin blockchain in an efficient manner. In the site regulation (SR) scenario, Bitcoin miners in the coal-based area are persuaded and suggested to relocate to the hydro-rich area to take advantage of the relatively lower cost of surplus energy availability in the area due to factors such as rain season, which results in only 20% of

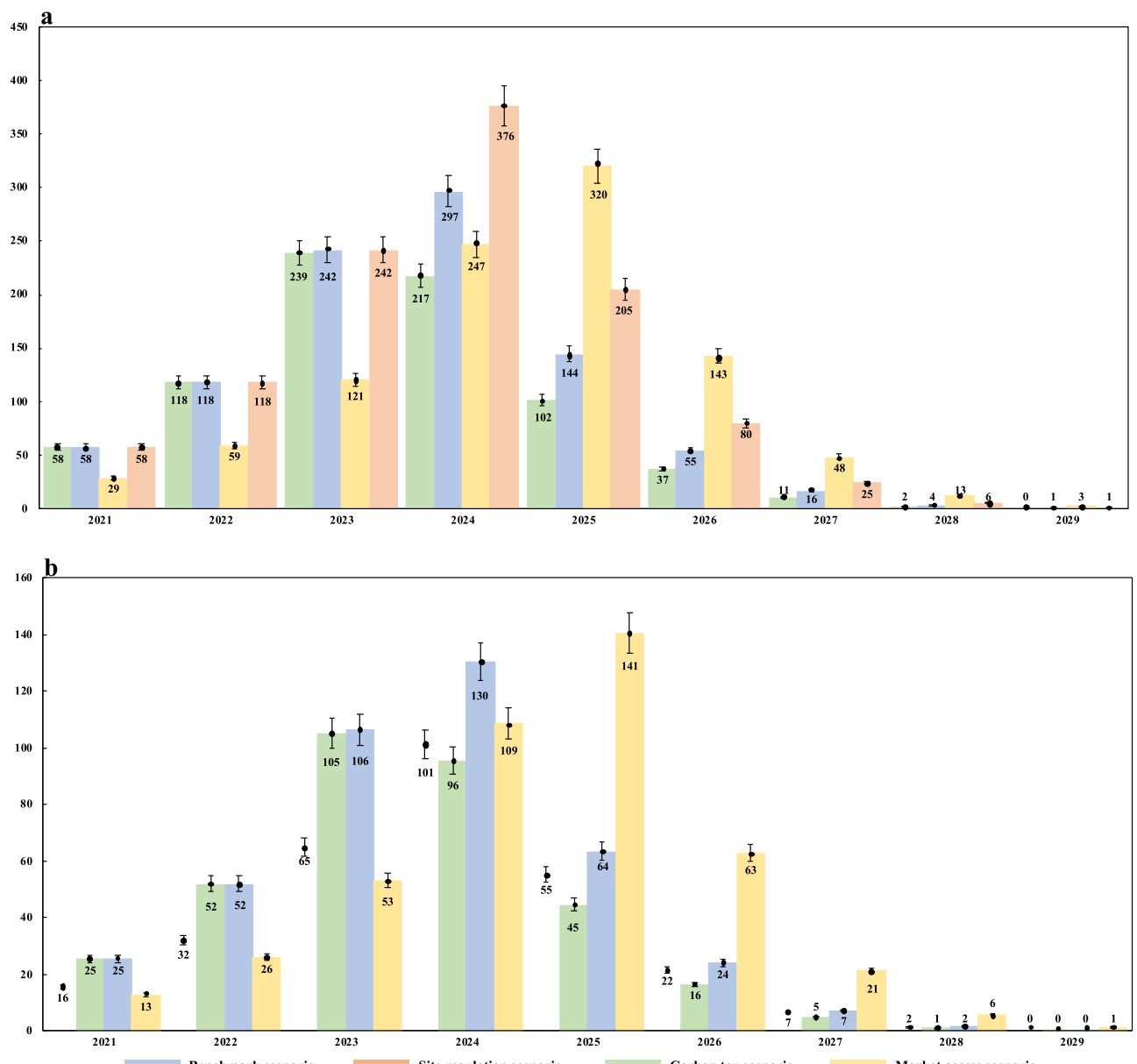

**Fig. 2 Estimated annualized scenario simulation results.** Estimated annualized energy consumption (**a**) and carbon emission flows (**b**) of Bitcoin operation in China are generated through monthly simulation results of BBCE modeling from 2021 to 2029. The blue, red, yellow, and green bars in **a** and **b** indicate the annual energy consumption and carbon emission flows of Chinese Bitcoin industry in benchmark, site regulation, market access, and carbon tax scenario, respectively. Each data is presented as mean values ± SEM based on 95% confidence intervals calculated by two-tailed *t*-tests (*p* < 0.05). *n* = 204 emission observations.

miners remaining in the coal-based area in the scenario. In the carbon tax (CT) scenario, carbon tax is increased to two-times the initial value to enforce more strict punishment for high carbon emission behaviors of Bitcoin blockchain. Utilizing the above scenarios, carbon emission flows and energy consumptions of Bitcoin blockchain are assessed, the carbon and energy reduction effectiveness of different policies are evaluated in BBCE simulations from the period of 2014–2030.

**Carbon emission flows of Bitcoin blockchain operation.** Without any policy interventions, the carbon emission pattern of the Bitcoin blockchain will become a non-negligible barrier against the sustainability efforts of China. The peak annual energy consumption and carbon emission of the Bitcoin blockchain in

China are expected to exceed those of some developed countries such as Italy, the Netherlands, Spain, and Czech Republic. Figure 2 reports the estimated annualized energy consumption and carbon emission flows of Bitcoin blockchain in China. As the baseline assessment under minimal policy intervention, the Benchmark scenario simulates the natural operation results of the Bitcoin blockchain. In the BM scenario, the annual energy consumption of Bitcoin blockchain in China will gradually grow and eventually peak in 2024, at 296.59 Twh per year. This suggests that Bitcoin industry operation would follow an energy intensive pattern. In fact, energy consumed by Chinese Bitcoin blockchain in 2024 will exceed the energy consumption level of Italy and Saudi Arabia in 2016, ranking it 12th among all the countries. Regarding the carbon tax scenario, the highest energy demand of the Bitcoin industry slightly decreases due to carbon emission

penalties, at 217.37 Twh. However, the results of the market access and site regulation scenarios indicate that the total energy consumption of the Bitcoin industry will reach 350.11 Twh and 319.80 Twh, respectively, in 2024 and 2025.

It is clear that the carbon emission behavior of the Bitcoin industry is consistent with the Bitcoin blockchain energy consumption intensity. In the BM scenario, annual carbon emission of the Bitcoin industry is expected to reach its maximum in 2024, at 130.50 million metric tons. In essence, the carbon emission pattern of the Bitcoin industry would become an increasing threat to China's greenhouse emission reduction target. At the international level, the estimated Bitcoin carbon emission in China exceeds the total greenhouse emission of the Czech Republic and Qatar in 2016, ranking it 36[th] worldwide. At the domestic level, the emission output of the Bitcoin mining industry would rank in the top 10 among 182 Chinese prefecture-level cities and 42 major industrial sectors. In comparison, the carbon emissions generated by Bitcoin blockchain experienced a significant reduction in SR and CT scenarios, which illustrate the positive impact of these carbon-related policies. On the contrary, the MA scenario witnesses a considerable increase of Bitcoin carbon emission to 140.71 million metric tons in 2025.

Based on the scenario results of the BBCE model, the Benchmark scenario indicates that the energy consumed and the carbon emissions generated by Bitcoin industry operation are simulated to grow continuously as long as mining Bitcoin maintains its profitability in China. This is mainly due to the positive feedback loop of the PoW competitive mechanism, which requires advanced and high energy-consuming mining hardware for Bitcoin miners in order to increase the probability of earning block rewards. In addition, the flows and long-term trend of carbon emission simulated by the proposed system dynamics model are consistent with several previous estimations[10,13], which are devoted to precisely estimate the carbon footprint of Bitcoin blockchain.

The Paris Agreement is a worldwide agreement committed to limit the increase of global average temperature[22,23]. Under the Paris Agreement, China is devoted to cut down 60% of the carbon emission per GDP by 2030 based on that of 2005. However, according to the simulation results of the BBCE model, we find that the carbon emission pattern of Bitcoin blockchain will become a potential barrier against the emission reduction target of China. As shown in Fig. 3, the peak annualized emission output of the Bitcoin mining industry would make it the 10[th] largest emitting sector out of a total of 42 major Chinese industrial sectors. In particular, it would account for approximately 5.41% of the emissions of the electricity generation in China according to the China Emission Accounts & Datasets (www.ceads.net). The peak carbon emission per GDP of Bitcoin industry is expected to sit at 10.77 kg per USD. In addition, in the current national economy and carbon emission accounting of China, the operation of the Bitcoin blockchain is not listed as an independent department for carbon emissions and productivity calculation. This adds difficulty for policy makers to monitor the actual behaviors of the Bitcoin industry and design well-directed policies. In fact, the energy consumption per transaction of Bitcoin network is larger than numerous mainstream financial transaction channels[17]. To address this issue, we suggest policy makers to set up separated accounts for the Bitcoin industry in order to better manage and control its carbon emission behaviors in China.

**Carbon policy effectiveness evaluation.** Policies that induce changes in the energy consumption structure of the mining activities may be more effective than intuitive punitive measures in limiting the total amount of energy consumption and carbon emission in the Bitcoin blockchain operation. Figure 4 presents the values of key parameters simulated by BBCE model. The carbon emission per GDP of the BM scenario in China is larger than that of all other scenarios throughout the whole simulation period, reaching a maximum of 10.77 kg per USD in June 2026. However, we find that the policy effectiveness under the MA and CT scenario is rather limited on carbon emission intensity reduction, i.e., the policy effectiveness of Market access is expected to reduce in August 2027 and that of Carbon tax is expected to be effective until July 2024. Among all the intended policies, Site Regulation shows the best effectiveness, reducing the peak carbon emission per GDP of the Bitcoin industry to 6 kg per USD. Overall, the carbon emission per GDP of the Bitcoin industry far exceeds the average industrial carbon intensity of China, which indicates that Bitcoin blockchain operation is a highly carbon-intense industry.

In the BM scenario, Bitcoin miners' profit rate are expected to drop to zero in April 2024, which suggests that the Bitcoin miners will gradually stop mining in China and relocate their operation elsewhere. However, it is important to note that the entire relocation process does not occur immediately. Miners with higher sunk costs tend to stay in operation longer than those with lower sunk costs, hoping to eventually make a profit again. Consequently, the overall energy consumption associated with Bitcoin mining remains positive until the end of 2030, at which time almost all miners would have relocated elsewhere. Correspondingly, the network hash rate is computed to reach 1775 EH per second in the BM scenario and the miner total cost to reach a maximum of 1268 million dollars. Comparing the scenario results for the three policies, the profitability of mining Bitcoin in China is expected to deteriorate more quickly in the CT scenario. On the other hand, Bitcoin blockchain can maintain profitability for a longer period in MA and SR scenarios.

Some attractive conclusions can be drawn based on the results of BBCE simulation: although the MA scenario enhances the market access standard to increase Bitcoin miners' efficiencies, it actually raises, rather than reduces, the emission output based on the simulation outcome. In the MA scenario, we observe the phenomenon of incentive effects proposed by previous works, which is identified in other fields of industrial policies, such as monetary policies, transportation regulations, and firm investment strategies[24–26]. In essence, the purpose of the market access policy is to limit the mining operations of low-efficiency Bitcoin miners in China. However, the surviving miners are all devoted to squeezing more proportion of the network hash rate, which enables them to stay profitable for a longer period. In addition, the Bitcoin industry in China generates more $CO_2$ emissions under the MA scenario, which can be mainly attributed to the Proof-of-Work (PoW) algorithm and profit-pursuit behaviors of Bitcoin miners. The results of the MA scenario indicate that market-related policy is likely to be less effective in dealing with high carbon emission behaviors of the Bitcoin blockchain operation.

The carbon taxation policy is widely acknowledged as the most effective and most commonly implemented policy on carbon emission reduction[27]. However, the simulation results of the CT scenario indicate that carbon tax only provides limited effectiveness for the Bitcoin industry. The carbon emission patterns of the CT scenario are consistent with the BM scenario until Bitcoin miners are aware that their mining profits are affected by the punitive carbon tax on Bitcoin mining. On the contrary, the evidence from the SR scenario shows that it is able to provide a negative feedback for the carbon emissions of Bitcoin blockchain operation. In our simulation, the maximized carbon emission per GDP of the Bitcoin industry is halved in the SR scenario in comparison to that in the BM scenario. It is interesting to note that although the peak annualized energy consumption cost of the Bitcoin mining industry in the SR scenario is higher than that in the BM scenario, a significantly higher proportion of miners have

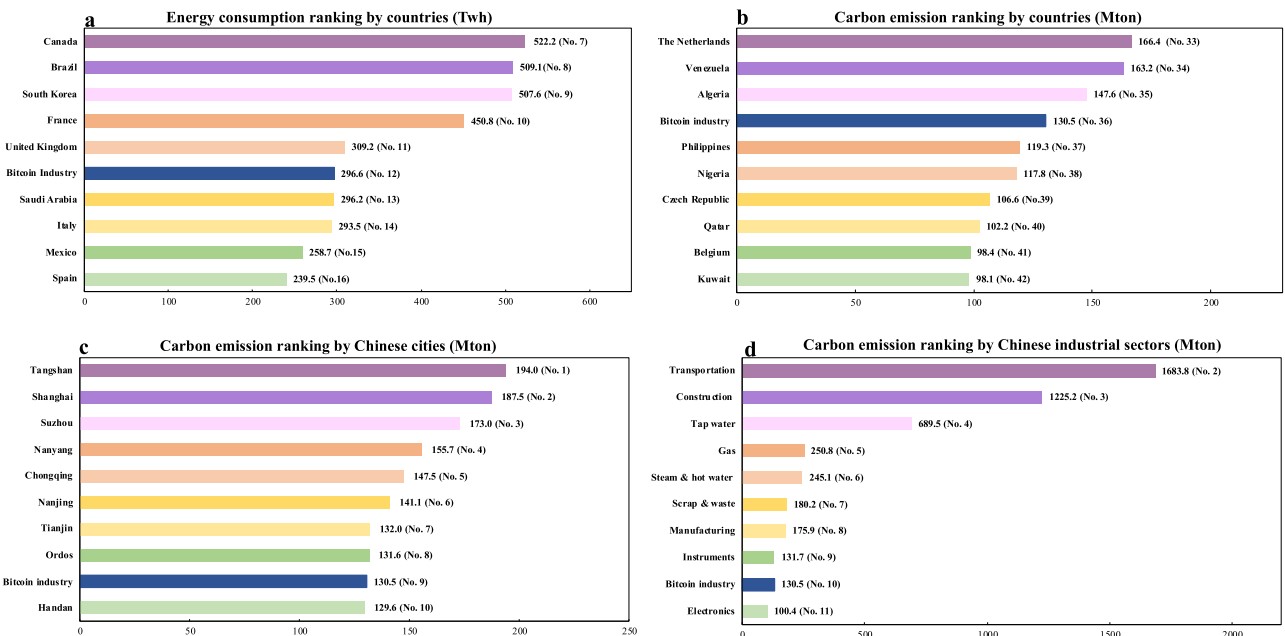

**Fig. 3 Bitcoin industry energy consumption and carbon emission comparisons.** In Fig. 3, the peak energy consumption and carbon emission of Bitcoin industry are compared to national level emissions of other countries as well as to the emissions of domestic cities and industrial sectors in China. Annual energy consumption and ranking by countries **a** are obtained from cia.gov (www.cia.gov), carbon emission and ranking by countries **b** are collected from global carbonatlas (www.globalcarbonatlas.org). The carbon emission by Chinese cities (**c**) and industrial sectors (**d**) are obtained from China Emission Accounts and Datasets (www.ceads.net). Due to the unreleased or missing data in some database, the above energy consumption and carbon emission data are obtained for 2016 level.

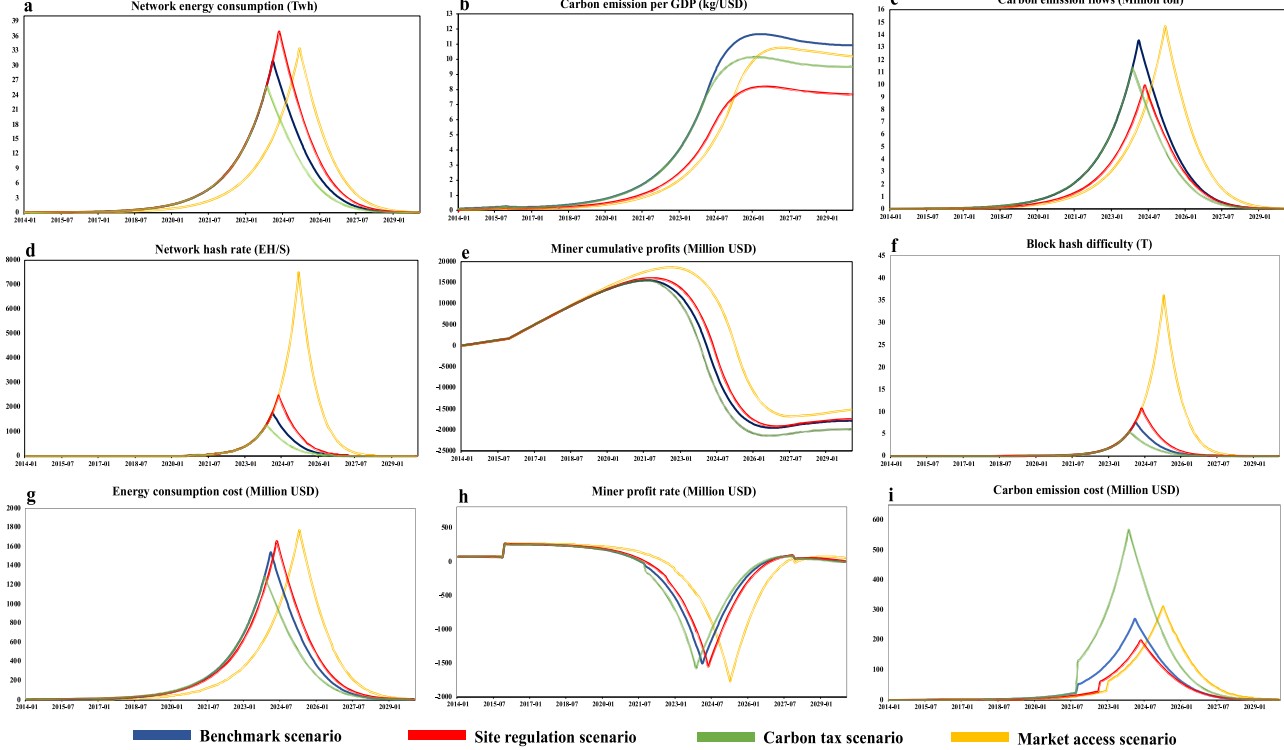

**Fig. 4 BBCE scenario assessment comparisons. a–i** Monthly network energy consumption (**a**), carbon emission per GDP (**b**), carbon emission flows (**c**), network hash rate (**d**), miner cumulative profits (**e**), block hash difficulty (**f**), energy consumption cost (**g**), miner profit rate (**h**), and carbon emission cost (**i**) under each intended policy are simulated and calculated by BBCE framework. Based on the regressed parameters of the BBCE model, the whole sample timesteps of network carbon emission assessment cover the period from January 2014 to January 2030.

relocated to conduct Bitcoin mining operation in the hydro-rich area in the SR scenario. Consequently, this naturally lowers the associated carbon emission cost in comparison to the BM scenario.

In general, the carbon emission intensity of the Bitcoin blockchain still far exceeds the average industrial emission intensity of China under different policy interventions, including limiting Bitcoin mining access, altering the miner energy consumption structure and implementing carbon emissions tax. This result indicates the stable high carbon emission property of Bitcoin blockchain operations. Nevertheless, it is rather surprising to arrive at the conclusion that the newly introduced cryptocurrency based on disruptive blockchain technology is expected to become an energy and carbon-intensive industry in the near future.

## Discussion

The current Proof-of-Work consensus algorithm used in the Bitcoin blockchain can potentially undermine the wide implementation and the operational sustainability of the disruptive blockchain technology. Overall, Bitcoin is a typical and pioneering implementation of blockchain technology. Its decentralized transaction characteristics and consensus algorithm provide a novel solution for trust mechanism construction, which can be beneficial and innovative for a variety of industrial development and remote transactions. In recent years, blockchain technology has been introduced and adopted by abundant traditional industries which seek to optimize their operation process in the real world[28], such as supply chain finance[29], smart contract[30], international business and trade[31], as well as manufacturing operations[32]. In addition, a national digital currency based on blockchain technology, namely Digital Currency Electronic Payment (DCEP), is scheduled and designed by The People's Bank of China, which is expected to replace the current paper-currency-based M0 supply in China.

However, the current consensus algorithm of Bitcoin, namely Proof-of-Work, gives rise to the hash rate competitions among Bitcoin miners for its potential block reward, which attracts an increasing number of miners to engage in an arms race and raise the energy consumption volumes of the whole Bitcoin blockchain. As a result, although PoW is designed to decentralize Bitcoin transactions and prevent inflation, we find that it would become an energy and carbon-intensive protocol, which eventually leads to the high carbon emission patterns of Bitcoin blockchain operation in China. The evidence of Bitcoin blockchain operation suggests that with the broaden usages and applications of blockchain technology, new protocols should be designed and scheduled in an environmentally friendly manner. This change is necessary to ensure the sustainability of the network—after all, no one wants to witness a disruptive and promising technique to become a carbon-intensive technology that hinders the carbon emission reduction efforts around the world. The auditable and decentralized transaction properties of blockchain provide a novel solution for trust mechanism construction, which can be beneficial and innovative for a variety of industrial development and remote transactions. However, the high GHG emission behavior of Bitcoin blockchain may pose a barrier to the worldwide effort on GHG emission management in the near future. As a result, the above tradeoff is worthy of future exploration and investigation.

Different from traditional industries, the carbon emission flows of emerging industries such as Bitcoin blockchain operation are unaccounted for in the current GDP and carbon emissions calculations. Without proper accounting and regulation, it is rather challenging to assess the carbon emission flows of these new industries using traditional tools such as input–output analysis. Through system dynamics modeling, our analysis constructs the emission feedback loops as well as captures the carbon emission

patterns. Furthermore, we are able to conduct emission assessment and evaluate the effectiveness of various potential implementable policies. Through scenario analysis, we show that moving away from the current punitive carbon tax policy consensus to a site regulation (SR) policy which induces changes in the energy consumption structure of the mining activities is more effective in limiting the total amount of carbon emission of Bitcoin blockchain operation. Overall, our results have demonstrated that system dynamics modeling is a promising approach to investigate the carbon flow mechanisms in emerging industries.

At the same time, we acknowledge there exists some limitations to our study and outline future directions for research. First, to reflect the true designed fundamental value of Bitcoin as intended by Nakamoto, our model assumes that the long-term Bitcoin price is primarily influenced by halving mechanism of Bitcoin mining rewards and is subjected to a linear increase every time a reward halving occurs. While the historical average Bitcoin price between each reward halving occurrence has generally followed this pattern since 2014, it is extremely volatile in real market operation and is subjected to the influence of other factors such as investor expectations. Therefore, a degree of uncertainty remains as to whether the linearity price assumption would hold, particularly as the Bitcoin market continues to grow into the future. Furthermore, our site regulation (SR) scenario assumes no cost on miners from relocating to clean-energy-based regions. In reality, there may be certain costs associated with this action, such as transportation. Therefore, although our results suggest that a site regulation (SR) policy may be more effective that the current punitive carbon tax policy consensus in limiting the total amount of carbon emission of Bitcoin blockchain operations, it is important to note that these are simulations arising from system dynamics modeling and are limited by the assumptions above.

Second, the projected carbon emissions of Bitcoin blockchain operation related to electricity production depends on the source which is used for its generation. In all of except for the Site Regulation (SR) scenario, we do not consider the potential changes of the Chinese energy sector in the future, which implies that miners would predominantly operate in the coal-based area. While this is certainly true as the current electricity mix in China is heavily dominated by coal, a series of efforts to incentivise electricity production on the basis of renewable energy sources (www.iea.org) and policies to increase the price for electricity generated on the basis of coal have been implemented. Consequently, these renewable energy-related efforts and policies can potentially affect the electricity consumption and subsequently, the amount of related carbon emission generated from Bitcoin blockchain operation.

Third, it is important to note that although our results suggest that with the broaden usage and application, blockchain technology could become a carbon-intensive technology that hinders the carbon emission reduction efforts around the world, as with any prediction model, many unforeseeable uncertainties could happen in the future that could cause the reality to deviate from the prediction. While it is true the blockchain technology, and Bitcoin as one of its applications, is, and increasingly will play a significant role in the economy, ultimately, the choice of adopting and using this technology lies in the hands of humans. Consequently, we should carefully evaluate the trade-offs before applying this promising technology to a variety of industries.

## Methods

This paper constructs a BBCE model to investigate the feedback loops of Bitcoin blockchain and simulates the carbon emission flows of its operations in China. In view of the complexity of Bitcoin blockchain operation and carbon emission process, the BBCE modeling for Bitcoin carbon emission assessment is mainly based on the following assumptions: (1) The electricity consumption of the Bitcoin mining process mainly consists of two types of energy: coal-based energy and hydro-based energy. (2) Bitcoin price is extremely volatile in real market

operations, which is inappropriate for long-term assessment in the BBCE model. Referring to the historical Bitcoin price data, we assume that the long-term Bitcoin price is mainly affected by the halving mechanism of Bitcoin mining rewards. (3) Miners gradually stop or choose other destinations for mining if the Bitcoin mining process is no longer profitable in China. (4) Bitcoin policies are consistent with the overall carbon emission flows in China. In other words, policies such as market access of Bitcoin miners and carbon tax of the Bitcoin blockchain operations can be rejiggered for different emission intensity levels. (5) Miners maintain full investment intensity while in operation, as any reduction in individual investment intensity would put miners in disadvantage and jeopardize their chances of mining new blocks and receiving the reward.

By investigating the inner feedback loops and causalities of the systems, BBCE modeling is able to capture the corresponding dynamic behaviors of system variables based on proposed scenarios[33,34]. Supplementary Fig. 1 shows the complete structure of BBCE modeling. The whole quantitative relationships of BBCE parameters are demonstrated in Supplementary Methods. Utilizing the flow diagram of BBCE systems illustrated in Supplementary Fig. 1, detailed feedback loops and flows of Bitcoin blockchain subsystems are discussed and clarified. The types, definitions, units, and related references of each variable in Supplementary Fig. 1 are reported in Supplementary Table 1.

**Bitcoin mining and transaction subsystem**. The Bitcoin blockchain utilizes Proof-of-Work (PoW) consensus algorithm for generating new blocks and validating transactions. Bitcoin miners earn a reward if the hash value of target blocks computed by their hardware is validated by all network participants. On the other hand, transactions packaged in the block are confirmed when the block is formally broadcasted to the Bitcoin blockchain. To increase the probability of mining a new block and getting rewarded, the mining hardware will be updated continuously and invested by network participants for higher hash rate, which would cause the hash rate of the whole network to rise. In order to maintain the constant 10-minute per new block generation process, the difficulty of generating a new block is adjusted based on the current hash rate of the whole Bitcoin network.

The halving mechanism of block reward is designed to control the total Bitcoin circulation (maximum of 21 million Bitcoins) and prevent inflation. Reward halving occurs every four years, which means that the reward of broadcasting a new block in Bitcoin blockchain will be zero in 2140. As a result, the Bitcoin market price increases periodically due to the halving mechanism of Bitcoin blockchain. With the growing popularity and broadened transaction scope of Bitcoin, the total transactions and transaction fee per block may steadily grow, which drive the other source of Bitcoin miner's profit rate. Overall, the profit of Bitcoin mining can be calculated by subtracting the total cost of energy consumption and carbon emissions from block reward and transaction fees. Miners will stop investing and updating mining hardware in China when the total cost exceeds the profit rate. Consequently, the whole network hash rate receives a negative feedback due to the investment intensity reductions.

**Bitcoin energy consumption subsystem**. The network mining power is determined by two factors: first, the network hash rate (hashes computed per second) positively accounts for the mining power increase in Bitcoin network when high hash rate miners are invested. However, the updated Bitcoin miners also attempt to reduce the energy consumption per hash, i.e., improve the efficiency of Bitcoin mining process, which helps to reduce the network mining power consumption. In addition, policy makers may raise the market access standard and create barriers for the low-efficiency miners to participate in Bitcoin mining activities in China. In terms of the energy consumption of the whole network, the power usage effectiveness is introduced to illustrate the energy consumption efficiency of Bitcoin blockchain as suggested by Stoll[13]. Finally, the network energy cost of Bitcoin mining process is determined by the network energy consumption and average electricity price, which further influences the dynamics behaviors of Bitcoin miner's investment.

**Bitcoin carbon emission subsystem**. The site selection strategies directly determine the energy types consumed by miners. Although the electricity cost of distinctive energies is more or less the same, their carbon emission patterns may vary significantly according to their respective carbon intensity index. In comparison to miners located in hydro-rich regions, miners located in coal-based regions generate more carbon emission flows under the similar mining techniques and energy usage efficiency due to the higher carbon intensity of coal-based energy[17]. The proposed BBCE model collects the carbon footprint of Bitcoin miners in both coal-based and hydro-based energy regions to formulate the overall carbon emission flows of the whole Bitcoin blockchain in China.

The level variable GDP consists of Bitcoin miner's profit rate and total cost, which suggests the productivity of the Bitcoin blockchain. It also serves as an auxiliary factor to generate the carbon emission per GDP in our model, which provides guidance for policy makers to implement punitive carbon taxation on Bitcoin industry. Finally, by combining both carbon cost and energy cost, the total cost of Bitcoin mining process provides a negative feedback for miner's profit rate and their investment strategies.

**BBCE model parameterizations and quantitative settings**. Our BBCE model has been constructed in Vensim software (PLE version 8.2.1). The time-related Bitcoin blockchain time-series data are obtained from www.btc.com, including network hash rate, block size, transaction fee, and difficulty. In addition, the auxiliary parameters and macroenvironment variables for network carbon emission flows assessment are set and considered through various guidelines. For example, the carbon intensities of different energies are suggested by Cheng et al.[35]. The average energy cost in China and carbon taxation are collected from the World Bank. The site proportion of Bitcoin miners in China are set based on the regional statistics of Bitcoin mining pools in www.btc.com. Moreover, the monthly historical data of Bitcoin blockchain are utilized for time-related parameter regression and simulation from the period of January 2014 to January 2020 through Stata software (version 14.1). Based on the regressed parameters, the whole sample timesteps of network carbon emission assessment cover the period from January 2014 to January 2030 in this study, which is available for scenario investigations under different Bitcoin policies. The initial value of static parameters in BBCE model are shown in Supplementary Table 2, the actual values of the parameterizations adopted are reported in Supplementary Methods, and the key quantitative settings of each subsystem are, respectively, run as follows:

According to the guidance of the Cambridge Bitcoin Electricity Consumption Index (https://www.cbeci.org) and Küfeoğlu and Özkuran[16], Bitcoin mining equipment is required to update and invest for remaining profitability. It is clear that mining hardware in the Bitcoin network consists of various equipments and their specifications. As a result, the investment intensity in Bitcoin blockchain is computed by the average price of a profitable mining hardware portfolio. The quantitative relationship between investment intensity and time can be expressed as the following form:

$$\text{Investment intensity} = \alpha_1 \times \text{Time} \times \text{Proportion of Chinese miners} \qquad (1)$$

In Eq. (1), the parameter $\alpha_1$ serves as the investment intensity function coefficient on time and the proportion of Chinese miners, which is estimated and formulated by the historical data of Bitcoin blockchain operation from the period of January 2014–January 2020. Then the Bitcoin miner profits are accumulated by profit rate and investment intensity flows, which can be obtained as follows:

$$\text{Miner cumulative profits}_t = \int_0^t (\text{Miner profit rate} - \text{Investment intensity}) dt \qquad (2)$$

As discussed above, the aim of Bitcoin mining hardware investment is to improve the miner's hash rate and the probability of broadcasting a new block. Utilizing the statistics of Bitcoin blockchain, the hash rate of the Bitcoin network is regressed, and the equation is:

$$\text{Mining hash rate} = e^{\beta_1 + \alpha_2 \text{Investment intensity}} \qquad (3)$$

Where $\beta_1$ and $\alpha_2$ represent the network hash rate constant function coefficient and coefficient on investment intensity, respectively. Similarly, the average block size of Bitcoin is consistent with time due to the growing popularity of Bitcoin transactions and investment. The block size is estimated by time and is illustrated as below:

$$\text{Block size} = e^{\beta_2 + \alpha_3 \text{Time}} \qquad (4)$$

Where $\beta_2$ and $\alpha_3$ indicate the block size function constant coefficient and coefficient on time, respectively. The proportion of Chinese miners in the Bitcoin mining process will gradually decrease if mining Bitcoin in China is not profitable. So, the proportion parameter in the BBCE model is set as follows:

$$\text{Proportion of Chinese miners} = \text{IF THEN ELSE(Miner cumulative Profits}$$
$$< 0, 0.7 - 0.01 \times \text{Time}, \ 0.7) \qquad (5)$$

Suggested by the mining pool statistics obtained from BTC.com, China accounts for approximately 70% of Bitcoin blockchain operation around the world. As a result, we set the initial proportion of Chinese Bitcoin miners as 70%. In addition, the proportion of Chinese Bitcoin miners will gradually decrease if the Bitcoin mining process is no longer profitable in China.

The energy consumed per hash will reduce, i.e., the mining efficiency of the Bitcoin blockchain will improve, when updated Bitcoin hardware is invested and introduced. Moreover, the market access standard for efficiency proposed by policy makers also affects network efficiency. Consequently, the mining efficiency can be calculated as follows:

$$\text{Mining efficiency} = e^{\beta_3 + \alpha_4 \times \text{Investment intensity} \times \text{Market assess standard for efficiency}} \qquad (6)$$

Where $\beta_3$ and $\alpha_4$ act as the mining efficiency function constant coefficient and coefficient on investment intensity and market access standard for efficiency, respectively. The above function coefficients of BBCE parameters are regressed and formulated based on the actual Bitcoin blockchain operation data from the period of January 2014 to January 2020, and the specific value of each parameter is reported in Supplementary Methods.

The mining power of the Bitcoin blockchain can be obtained by network hash rate and mining efficiency. The equation of mining power is shown as follows:

$$\text{Mining power} = \text{Mining hash rate} \times \text{Mining efficiency} \qquad (7)$$

Finally, the energy consumed by the whole Bitcoin blockchain can be expressed by mining power and power usage effectiveness:

$$\text{Network energy consumption} = \text{Mining power} \times \text{Power usage effectiveness} \quad (8)$$

Employing the regional data of Bitcoin mining pools, coal-based and hydro-based energy is proportionally consumed by distinctive Bitcoin pools. The total carbon flows in Bitcoin blockchain are measured by the sum of both monthly coal-based and hydro-based energy carbon emission growth. The integration of total carbon emission is:

$$\text{Total carbon emission}_t = \int_0^t \text{Carbon emission flow } dt \quad (9)$$

In addition, carbon emissions per GDP are introduced to investigate the overall carbon intensity of the Bitcoin mining process in China, which is formulated by the following equation:

$$\text{Carbon emission per GDP} = \text{Carbon emission/GDP} \quad (10)$$

Suggested by the World Bank database, we introduce the average taxation percentage for industrial carbon emission (1%) as the initial carbon tax parameter in BBCE modeling. In addition, the punitive carbon taxation on the Bitcoin blockchain will be conducted by policy makers, i.e, the carbon taxation on the Bitcoin blockchain will be doubled, if the carbon emission per GDP of the Bitcoin blockchain is larger than average industrial carbon emission per GDP in China (2 kg/GDP). As a result, the carbon tax of Bitcoin blockchain is set as:

$$\text{Carbon tax} = 0.01 \times \text{IF THEN ELSE(carbon emission per GDP} > 2, 2, 1) \quad (11)$$

**Validation and robustness tests**. In order to test the suitability and robustness of BBCE modeling system structures and behaviors, three model validation experiments are introduced and conducted in our study, i.e., the structural suitability tests (see Supplementary Fig. 3), reality and statistical tests (see Supplementary Fig. 4), and sensitivity analysis (see Supplementary Fig. 5). The validation results of the three tests are reported in Supplementary Discussion. Overall, the model validation results indicate that the proposed BBCE model can effectively simulate the causal relationship and feedback loops of carbon emission system in Bitcoin industry, and the parameters in BBCE model have significant consistencies with actual Bitcoin operating time-series data. In addition, the sensitivity analysis of BBCE model also shows that a slight variation of the BBCE parameters does not lead to the remarkable changes in the model behaviors or the ranking of the intended carbon reduction policies, thus indicating that the proposed BBCE model has excellent behavioral robustness and stability.

**Reporting summary**. Further information on experimental design is available in the Nature Research Reporting Summary linked to this paper.

## Data availability
All original datasets used and the generated data from the results of the study are available at CEADS database (https://www.ceads.net/user/download-anonymous.php?id=1083). All data are also available from the corresponding authors upon reasonable request.

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

## Acknowledgements
This work was supported by grants from the National Natural Science Foundation of China (71988101 and 72022019).

## Author contributions
S.J. and Y.L. contributed to conceptualizing and designing the work, acquiring the data, conducting the analysis, interpreting the data, writing, and revising of the paper. Q.L. interpreted the data. Y.H. revised the paper. D.G. revised the paper and supervised the work. Y.X. revised the paper. S.W. conceptualized the work, revised the paper, and supervised the work.

## Competing interests
The authors declare no competing interests.
