## [Peer Review File · Nature Communications]

REVIEWER COMMENTS

Reviewer #1 (Remarks to the Author):

The article aims to outline the effect of different policies on the carbon emission flow of Bitcoin blockchain transactions in China by applying a Bitcoin Blockchain Carbon Emission Model. The topic of the article is highly relevant as it combines the emergence of blockchain technology applications with the global challenge of climate change mitigation, i.e. emission reduction from the energy sector.

Policy interventions are highly relevant in order to reduce the amount of emissions caused by the energy consumption of Bitcoin blockchain transactions. As a first step it is thus relevant to understand the implications of different policies. This article attempts to contribute to this knowledge gap.

Generally, the article is well structured and paragraphs are organised logically. Graphs and illustrations are very clear and a useful addition to the text.

Major comments:

The main comment concerns the lacking explanation of limitations of this research. This concerns mainly the applied model.

The model aims at incorporating a variety of factors, however, a critical reflection on whether and to which extent these factors are assumptions prone to be influenced by external circumstances is lacking. Such an explanation would be helpful in order to understand the resilience of the model and thus its usefulness for its application for the development of policies. For example, line 404ff presents the assumption that the price of Bitcoin is primarily influenced by the reduction (halving) of units. While this is theoretically true, the assumption is built on linear price expectations. While this works for the model, it would at least be necessary to critically reflect on this assumption, for example by presenting price development data from the past which would provide evidence whether and to which extent the linearity assumption is realistic, or not.

Furthermore, the article seems to apply a strong technology determinism perspective. While it is true that blockchain technology, and Bitcoin as one of its applications, is, and increasingly will play a significant role in the economy, it does not determine society. Again, this might relate to the lacking reflection on the limitations of the model.

Another limitation that should be further elaborated is the assumptions concerning the Chinese energy sector. The authors present the peak electricity consumption of Bitcoin blockchain operations in 2024 and subsequently identify the related carbon emissions. However, it should be made clear that the amount of emissions related to electricity production depends on the source which is used for its generation. Certainly, the electricity mix in China is heavily dominated by coal, yet, efforts to incentivise electricity production on the basis of renewable energy sources are launched (see International Energy Agency, country reports, China). Moreover, China introduced an emission trading scheme (ETS) which is in force since 2020. This ETS mainly covers coal- and gas-fired powerplants and, ideally, will increase the price for electricity generated on the basis of coal or gas. The article would gain by information presenting some background information on the energy mix and existing regulation on reducing emissions from the energy sector in China. In this context the article should also be revised concerning the use of the terms "energy" and "electricity" which seem to be mixed up sometimes.

Minor points:

Typos: Line 59: "The network mining power in is", line 82 "Netherlands" should be "the Netherlands"
line 99: assesses, line 175: access (instead of assess), line 215: network, line 259: PoW abbreviation needs to be explained earlier

Explanations: Line 66: "heavy and clean energy regions": needs to be explained

Suggestion: the final paragraph (line 312-320) could be integrated in the introduction as the information is quite basic and thus seems to be out of place at the very end of the article.

Reviewer #2 (Remarks to the Author):

The paper presents an interesting study on the environmental impact of bitcoin industry growth in China. There are some points to consider, revise and improve throughout the paper, before discussing the insights that model results convey.

Abstract

Line 11: clarify this study involves a simulation model

Line 20: the incentive effect is cryptic. Instead of incentive clarify whether it is a positive or negative effect.

Remarks on the content

1. In several points in the text, bitcoin GHG emissions are compared to national level emissions of other countries. While this is useful in conveying a general sense of scale to the reader, it would be better to express emissions as a percentage of total GHG emissions in China, or alternatively, the total annual emissions of the electricity generation sector in China. This will show how much of an impact bitcoin has on Chinese ambitions to meet the Paris agreement, and therefore the urgency, and magnitude of policies to reduce them.

2. Figure 3d. A rapidly decreasing profit rate, should slow down bitcoin investments and this in turn should slow down the decrease in profit rate until it reaches a value close to zero. Instead what Figure 3d shows is that profits appear to reach a stable negative value i.e. losses, an unsustainable state for any investor for a long period of time. Do losses persist in the case when the model is simulated to 2050?

3. Figure 4. introduce plus and minus signs in line with standard system dynamics notation (Lane, 2000). Replace heavy energy with Coal based energy in all related variables. Replace Clean energy with hydro based energy. In general, strive to make variable names more specific and accurate as to what they represent in reality.

Lane DC. 2000. Diagramming conventions in system dynamics. Journal of the Operational Research Society 51(2), 241-245.

5. Furthermore use more meaningful variable names. For example, proportion of what? same for difficulty, efficiency, market access (standard?) and so on.

6. State whether Figure 4 shows the complete structure of the model or a more aggregate/simplified version of it. This clarification should be made because it is not clear how the number of people involved in bitcoin investments grows due their attractiveness. In Figure 4, it appears that this is included somehow in variable Hash rate but there is no equation for it in the manuscript.

7. Regarding equations 1-6: clarify what α , β are. Use meaningful variable names. e.g. proportion or efficiency is uninformative.

8. All variable names should be consistent with Figure 4.

9. What does 0.7 represent in equation (5)? Why is this parameter value chosen?

10. Provide the rationale for equation 11, what do the values 0.01, 1, 2 represent?

11. The model has been constructed in vensim software.

12. To improve model transparency, it is standard practice to submit the model documentation using the SDM tool <https://www.systemdynamics.org/SDM-doc>

13. In the revised version submit the output of the SDM tool as supplementary material.

14. Line 570: you refer to the results of model validation. Provide the graphs, on hash rate and efficiency.

Text remarks

Line 35: takes

Line 42: capture and reproduce the endogenous dynamics of complex system elements (Sterman, 2000; Richardson, 2011)

Line 62: From the text it is evident that PUE should stand for power usage efficiency.

Line 96: assesses
Line 128: replace closeness with proximity
Line 132: replace with: China is a key signatory of the Paris agreement
Line 144: start sentence with: As suggested...
Line 149: market access standard for efficiency
Line 151: what does matian mean? Replace with another word.
Line 169: replace maximize with peak
Figure 2: change the format to that of figure 3 for consistency
Line 249: replace attracting with attractive
Line 251: this is an obscure sentence. What does emissions prompted policy mean? Replace emission reduced with emission reduction.
Line 256: the surviving miners
Line 258: the bitcoin industry in China generates more...

Reviewer #3 (Remarks to the Author):

The main novelty proposed is to perform de-carbonisation policy experiments on a model of Bitcoin mining carbon emissions in China (BBCE), based on system dynamics. According to BTC.com, as of today, China accounts for 78% of worldwide Bitcoin mining activity. With a model, the current status quo (BR or benchmark scenario) can be counter-factually assessed quantitatively against alternative policy scenarios: increasing the carbon tax (CT scenario), restricting market access for half of the current miners (MA scenario) or halving the proportion of miners currently using 'dirty' power sources (SR scenario). The main novelty of the model-based approach is that if nothing is done, carbon emissions are expected to curb on their own from 2024 onwards (peaking then at 296.59TWh per year). Instead, they are expected to peak in 2023 at 217.37TWh when increasing the carbon tax from 2 to 5% (CT scenario), in 2024 (at 350.11 TWh) under scenario SR and in 2025 (at 319.80 TWh) under scenario MA. The corresponding carbon emissions at their maximum correspondingly rank as follows: SR (not provided, in Figure 2 panel B) < CT (not provided, in Figure 2 panel B) < BR (130.5 MtCO₂) < MA (140.71 MtCO₂). The authors then conclude that restricting Bitcoin miners' use of 'dirty' sources of energy (e.g. coal-based) appears more policy effective than increasing the carbon tax, particularly in the striking simulated reduction of carbon emissions per USD from 10.77kg (BR scenario) to 6kg (SR scenario), cf. Figure 3. Based on the fact that, currently, emissions from Bitcoin mining are not officially accounted for, the authors further suggest their official inclusion to avoid jeopardizing China's stated intentions to cut down emissions by 2030 to 60% of their level in 2005, to have a chance of meeting the Paris Agreement target.

SD-modelling the carbon emissions associated with Bitcoin mining introduces 'self-correcting' feedback loops, that 'curb' greenhouse emissions naturally due to the 'competitive effect' associated with the proof of work algorithm, i.e. when either miners' rewards fall (e.g. Bitcoin price, or transaction fees) or miners' costs increase (e.g. increased competition between miners will increase the overall network hashrate, making it more difficult to successfully find a block). In the absence of policy interventions, the model forecasts a 'peak' in power consumption, and hence in associated carbon emissions. Experimenting with alternative policy interventions may bring that peak forwards or backwards, as well as entail changes in the level of emissions per USD relative to the benchmark scenario (BR). But for the exercise to be of use and its policy-based predictions credible, the model needs to be (i) clearly understood, (ii) adequately validated, and (iii) robust (e.g. some sensitivity analysis on the main model parameters gives us a notion of how reliable the model predictions, and policy exercises conducted on its basis, are). Ideally, one would also need some notion of (iv) 'reliability' in terms of how confident one can be on the model predictions and policy evaluation exercises, i.e. some confidence intervals around model-based point estimates/results from alternative policy scenarios. I comment more on each of those points below:

(iv) Model clarity: The verbal description of the SD modelling of Bitcoin Blockchain Carbon Emissions (BBCE) does not allow the reader to understand the contribution relative to the existing literature: e.g. (a) GDP does not measure productivity, but measures based on 'changes in GDP' like GDP growth, does. (b) The allowed feedback loops presumably give incentives to Bitcoin miners (through reduced incomes, approximated by GDP) to reduce investments in mining (e.g. upgrading antMiners) and/or mining efforts, everything else constant. However, that would imply a reduced power consumption, not an increased one (to reach the projected 296.59 TW/h by 2024, and generate 130.50 MtCO_{2e}), which is why the contribution is unclear (further Fig. 5 could not be found, and instead the reader assumed that Fig. 4 –Methods section-- provided a flow diagram of the SD model structure: please correct me if wrong. Appendix B provides the list of BBCE model equations). (c) The actual values of the parameterizations adopted are unreported, e.g. in equations (1), (3), (4), (6), etc. the values of the parameters $\alpha_1, 2, 3, 4$ and/or $\beta_1, 2, 3$ are never reported, reducing the reader's ability to comment on reasonableness and/or replicability. (d) Some recent research reports that it is important to understand seasonal movements of miners within China to take advantage of cheap surplus energy availability due to the rain season (e.g. Stoll et al., 2019). According to Fig. 4, this would correspond to introducing a causal arrow from either the electricity price or the electricity cost to miners' location decisions, which is currently absent. If I wanted to have a sense of the quantitative impact of this more realistic scenario on the reported authors' point predictions for emissions, it remains unclear from the current presentation efforts how to think about it? And it would certainly change the estimates obtained under scenario SR, in Table 1.

(v) Model validation is so synthetic that I could not understand, e.g. what does '0.9, at 0.977 and 0.913 respectively,' (in lines 570-1 of the manuscript) mean? The authors should provide at least the definition and critical values of the statistical tests used to come up with statements such as this, because model validation is crucial for rendering the policy evaluation exercises credible. Another way to validate the novel model predictions would be to circumscribe it to the time window for which recent research reports estimates of Bitcoin power consumption and associated carbon emissions (e.g. Stoll et al., 2019, report in a figure their estimates in relation to those obtained until 2018, at an annual frequency; CBECI provides an online tool against which the authors' model based results could be benchmarked, etc.). More concretely: In Appendix C it is stated that 'It is estimated that between the period of January 1st, 2016 and June 30th, 2018, up to 13 million metric tons of CO₂ emissions can be attributed to the Bitcoin Blockchain' which are certainly below the estimates reported in recent publications (e.g. Stoll et al., 2019 report about 22 MtCO₂ for 2018 alone). Yet, the SD-modelling departs from such low point estimates, and forecasts them to increase almost 25-fold by 2024?

(vi) Robustness of the model to alternative parameterizations, e.g. a power usage of electricity (PUE) of 1.10 is adopted citing Stoll et al. (2019), without further discussion. But one wonders how robust the reported estimates (and results on the ranking of the different policies considered) would be to more realistic measures of 1.05 or 1.0 (e.g. for pools in Inner Mongolia, where cooling is unnecessary). Similarly, one of the main sources of uncertainty in estimating carbon emissions, are the actual carbon intensities of different sources of electricity, and the reader wonders how robust the reported point estimates are to alternative carbon intensities (or alternative proportions of miners using 'clean' energy sources, in terms of scenario SR).

(ii) Reliability of reported point estimates: policy decision makers typically worry about the percentage chances that a given policy is going to have unintended consequences, i.e. about how confident the researcher is when recommending some policy change on the basis of her/his model. This is typically encapsulated in some notion of 'reliability' of the reported mean forecast or point estimate, as measured by their prediction intervals (PI) (e.g. with a 95% chance, emissions are not going to be higher than the upper bound of the PI, nor lower than the lower bound). The authors should provide some 'reliable bounds' associated with their point estimates (and results of their alternative policy scenarios), or explicitly warn the reader about the difficulty of doing so, providing a valid reason.

In relation to the above points, some stated claims need further clarification, like (line 234): 'In the BM scenario, Bitcoin miner profits are expected to drop to zero in April 2024, which suggests that the Bitcoin miners will gradually stop mining in China and relocate their operations elsewhere'. If that is the case, it is unclear how the overall energy consumption associated with Bitcoin mining can carry on

being positive between April 2024 and the end of 2030, as reported in the top-right panel of Figure 3, i.e. better understanding of the model, baseline model parameters and estimated regression parameters, is crucial to gauge whether the reported predictions make actual sense, lending further credibility to the policy experiments considered.

Some additional minor comments follow:

-Needs serious English proof-reading.

-Some references cited do not support the written sentence, e.g. (line 133) 'However, without appropriate interventions and feasible policies, the intensive Bitcoin blockchain operations in China can quickly grow as a threat that could potentially undermine the emission reduction effort taken place in the country¹⁰'

But then reference No. 10 refers to Ebola?

Redding, D. W., Atkinson, P. M., Cunningham, A. A., Iacono, G. L., Moses, L. M., Wood, J. L., & Jones, K. E. Impacts of environmental and socio-economic factors on emergence and epidemic potential of Ebola in Africa. *Nat. Commun.* 10, 1-11 (2019).

Signed: Dr. Hector F. Calvo-Pardo

Detailed Response to Reviewers

We would like to express our heartfelt gratitude to three anonymous reviewers for their valuable feedback. Our manuscript, titled “Policy assessments for the carbon emission flows and sustainability of Bitcoin blockchain operation in China”, benefited significantly from the constructive comments and insights from the review team. Based on the suggestions received, we have made careful revisions to the original manuscript. In the revised manuscript, all changes are marked in red. In addition, we have also carefully proofread this manuscript for typographical, grammatical, and other errors. We hope the revised manuscript is able to meet your standard of quality and address the concerns raised by the reviewers. In the following, you will find our detailed point-by-point responses to the reviewers’ comments.

1 Responses to Reviewer #1

We sincerely appreciate your valuable feedback on our manuscript. All of these comments have helped us significantly in improving the quality of our manuscript. Based on your comments and suggestions, we have made extensive changes to the manuscript in the revised manuscript. Our responses to your specific comments are provided below. We also provide a copy of the relevant sections (marked in red) following our responses.

Major Comments

Comment 1: The model aims at incorporating a variety of factors, however, a critical reflection on whether and to which extent these factors are assumptions prone to be influenced by external circumstances is lacking. Such an explanation would be helpful in order to understand the resilience of the model and thus its usefulness for its application for the development of policies. For example, line 404ff presents the assumption that the price of Bitcoin is primarily influenced by the reduction (halving) of units. While this is theoretically true, the assumption is built on linear price expectations. While this works for the model, it would at least be necessary to critically reflect on this assumption, for example by presenting price development data from the past which would provide evidence whether and to which extent the linearity assumption is realistic, or not.

Answer: Thank you for pointing this out. To address the lacking explanation of limitations, we have added discussions in the manuscript on whether and to which extent the factors incorporated in the model are assumptions prone to be influenced by external circumstances. In particular, we acknowledge that while the historical average Bitcoin price between each reward halving occurrence since 2014 has generally followed our linearity Bitcoin price assumption presented in line 404ff, its high volatility in real market operations and influence of other factors such as investor expectations does place a certain degree of uncertainty on whether this linear price assumption will hold, particularly as the Bitcoin market continues to grow into the future. The specific content for this change is as follows:

- Page 15, Line 342-350 (Discussion)

At the same time, we acknowledge there exists some limitations to our study and outline future directions for research. First, to reflect the true designed fundamental value of Bitcoin as intended by Nakamoto, our model assumes that the long-term Bitcoin price is primarily influenced by halving mechanism of Bitcoin mining rewards and is subjected to a linear increase everytime a reward halving occurs. While the historical average Bitcoin price between each reward halving occurrence has generally followed this pattern since 2014, it is extremely volatile in real market operations and is subjected to the influence of other factors such as investor expectations. Therefore, a degree of uncertainty remains as to whether the linearity price assumption would hold, particularly as the Bitcoin market continues to grow into the future.

Comment 2: Furthermore, the article seems to apply a strong technology determinism perspective. While it is true that blockchain technology, and Bitcoin as one of its applications, is, and increasingly will play a significant role in the economy, it does not determine society. Again, this might relate to the lacking reflection on the limitations of the model.

Answer: Thank you for this suggestion. We definitely agree with your view. Although our results do suggest that with the broaden usage of application, blockchain technology could become a carbon-intensive technology that hinders the carbon emission reduction efforts around the world, as with any prediction model, many unforeseeable uncertainties could happen in the future that could cause the reality to deviate from the prediction. Ultimately, the choice of adopting and using this technology lies in the hands of humans, so it does not determine the outcome of the society. We have pointed this out as part of our discussion. The specific content for this change is as follows:

- Page 16, Line 363-370 (Discussion)

Third, it is important to note that although our results suggest that with the broaden usage and application, blockchain technology could become a carbon-intensive technology that hinders the carbon emission reduction efforts around the world, as with any prediction model, many unforeseeable uncertainties could happen in the future that could cause the reality to deviate from the prediction. While it is true the blockchain technology, and Bitcoin as one of its applications,

is, and increasingly will play a significant role in the economy, ultimately, the choice of adopting and using this technology lies in the hands of humans. Consequently, we should carefully evaluate the trade-offs before applying this promising technology to a variety of industries.

Comment 3: Another limitation that should be further elaborated is the assumptions concerning the Chinese energy sector. The authors present the peak electricity consumption of Bitcoin blockchain operations in 2024 and subsequently identify the related carbon emissions. However, it should be made clear that the amount of emissions related to electricity production depends on the source which is used for its generation. Certainly, the electricity mix in China is heavily dominated by coal, yet, efforts to incentivise electricity production on the basis of renewable energy sources are launched (see International Energy Agency, country reports, China). Moreover, China introduced an emission trading scheme (ETS) which is in force since 2020. This ETS mainly covers coal- and gas-fired powerplants and, ideally, will increase the price for electricity generated on the basis of coal or gas. The article would gain by information presenting some background information on the energy mix and existing regulation on reducing emissions from the energy sector in China. In this context the article should also be revised concerning the use of the terms “energy” and “electricity” which seem to be mixed up sometimes.

Answer: Thank you for this suggestion. In all of except for the Site Regulation (SR) scenario, we do not capture the potential changes of the Chinese energy sector in the future, which implies that miners would predominantly operate in the coal-heavy area. This is certainly an assumption concerning the Chinese energy sector and part of the limitation of our research. Consequently, we have added some background information on the energy mix and existing regulation on reducing emission from the energy sector in China as part of our discussion. In addition, we have corrected all the mixed up use of the terms “energy” and “electricity” in the revised manuscript. To ensure consistency, the term “energy” is used throughout the revised manuscript. The specific content for this change is as follows:

- Page 15, Line 352-361 (Discussion)

Second, the projected amount of Bitcoin blockchain operations carbon emissions related to electricity production depends on the source which is used for its generation. In all of except for the Site Regulation (SR) scenario, we do not capture the potential changes of the Chinese energy sector in the future, which implies that miners would predominantly operate in the coal heavy area. While this is certainly true as the current electricity mix in China is heavily dominated by coal, a series of efforts to incentivise electricity production on the basis of renewable energy sources (IEA, China) and policies to increase the price for electricity generated on the basis of coal have been implemented. Consequently, these renewable energy-related efforts and policies can potentially affect the electricity consumption and subsequently, the amount of related carbon emission generated from Bitcoin blockchain operations.

Minor Comments

Comment 4: Typos: Line 59: “The network mining power in is”, line 82 “Netherlands” should be “the Netherlands”; line 99: assesses, line 175: access (instead of assess), line 215: network, line 259: PoW abbreviation needs to be explained earlier

Answer: Thank you for pointing this out. We sincerely apologize for the typos in our manuscript and have carefully corrected them in our revised manuscript. The specific changes are as follows:

- Page 3, Line 57

The network mining power is determined by two factors

- Page 3, Line 79-80

which exceeds the total energy consumption level of Italy and Saudi Arabia and ranks 12th among all countries in 2016.

- Page 8, Line 170-182

However, the results of the Market access and Site regulation scenarios indicate that the total energy consumption of the Bitcoin industry will reach 350.11 Twh and 319.80 Twh respectively in 2024 and 2025.

- Page 10, Line 226-227

In fact, the energy consumption per transaction of Bitcoin network is larger than numerous mainstream financial transactions channels

- Page 4, Line 96-97
the PoW(Proof-of-Work) consensus algorithm

Comment 5: Explanations: Line 66: “heavy and clean energy regions”: needs to be explained

Answer: Thank you for pointing this out. The heavy energy region in our model refers to the region where the main method of electricity generation is coal-based, while the clean energy region represents the region where the main method of electricity generation is hydro-based. We have included the additional explanation in the revised manuscript. The specific content for this change is as follows:

- Page 3, Line 63-65 (Introduction)

The BBCE model collects the carbon footprint of Bitcoin miners in both coal-based energy and hydro-based energy regions to formulate the overall carbon emission flows of the whole Bitcoin industry in China.

Comment 6: Suggestion: the final paragraph (line 312-320) could be integrated in the introduction as the information is quite basic and thus seems to be out of place at the very end of the article.

Answer: Thank you for this suggestion. We have integrated the final paragraph in the introduction in the revised manuscript. The specific change is follows:

- Page 2, Line 36-45 (Introduction)

In recent years, the system dynamics (SD) based model is widely introduced for carbon emission flows estimation for a specific area or industry^{4,5}. In comparison to its counterparts, SD modelling has two main advantages in carbon emission flows assessment: first, by combining the feedback loops of stock and flow parameters, system dynamics technique is able to capture and reproduce the endogenous dynamics of complex system elements, which enables the simulation

and estimation of specific industry operations^{6,7,8}. In addition, since the SD based model is focused on disequilibrium dynamics of the complex system^{9,10}, intended policies can be adjusted for scenario policy effectiveness evaluation. Consequently, based on system dynamics modeling, we develop the Bitcoin blockchain carbon emission model (BBCE) to assess the carbon emission flows of the Bitcoin network operations in China under different scenarios.

2 Responses to Reviewer #2

We are grateful for your valuable feedbacks on our manuscript. All of these comments have helped us significantly in improving the quality of our manuscript. Based on your comments and suggestions, we have made extensive changes to the manuscript in the revised manuscript. Our responses to your specific comments are provided below. We also provide a copy of the relevant sections (marked in red) following our responses.

Abstract

Comment 1: Line 11: clarify this study involves a simulation model

Answer: Thank you for pointing this out. We have clarified that this study involves a simulation model in the abstract of the revised manuscript. The specific content of the change is as follows:

- Page 1, Line 11-14 (Abstract)

By investigating the carbon emission flows of Bitcoin blockchain operations in China with a simulation-based Bitcoin blockchain carbon emission (BBCE) model, we find that without any policy interventions, the annual energy consumption of the Bitcoin blockchain in China is expected to peak in 2024 at 296.59 Twh and generate 130.50 million metric tons of carbon emission flows correspondingly.

Comment 2: Line 20: the incentive effect is cryptic. Instead of incentive clarify whether it is a positive or negative effect.

Answer: Thank you for pointing this out. We sincerely apologize for the cryptic term in our manuscript. The incentive effect is a negative effect. However, due to the word limit of the abstract, we have decided to not mention it in the abstract. Instead, we emphasize our finding that policies inducing changes in the energy consumption structure of the mining activities are more effective

than intuitive punitive measures in limiting the carbon emission of Bitcoin blockchain operations. The specific changes are as follows:

- Page 1, Line 18-20 (Abstract)

Through scenario analysis, we show that policies inducing changes in the energy consumption structure of the mining activities are more effective than intuitive punitive measures in limiting the total amount of carbon emission of Bitcoin blockchain operations.

Remarks on the content

Comment 3: In several points in the text, bitcoin GHG emissions are compared to national level emissions of other countries. While this is useful in conveying a general sense of scale to the reader, it would be better to express emissions as a percentage of total GHG emissions in China, or alternatively, the total annual emissions of the electricity generation sector in China. This will show how much of an impact bitcoin has on Chinese ambitions to meet the Paris agreement, and therefore the urgency, and magnitude of policies to reduce them.

Answer: Thank you for this suggestion. We certainly agree that conducting emission comparisons domestically within China would provide a clearer understanding on the impact bitcoin has on Chinese ambitions to meet the Paris agreement and the urgency to reduce it. According to the China Emission Accounts & Datasets (www.ceads.net), our estimated peak emission generated from bitcoin mining operations would account for approximately 5.41% of the total emissions of electricity generation in China. In addition to stating this result, we have added domestic comparisons in the revised manuscript, including comparisons with the emission output of 182 Chinese prefecture-level cities and 42 major industrial sectors. The specific contents for this change are as follows:

- Page 1, Line 13-18 (Abstract)

we find that without any policy interventions, the annual energy consumption of the Bitcoin blockchain in China is expected to peak in 2024 at 296.59 Twh and generate 130.50 million metric tons of carbon emission flows correspondingly. Internationally, this level of emission output would

exceed the total annualized greenhouse gas emission output of the Czech Republic and Qatar. Domestically, it ranks in the top 10 among 182 prefecture-level cities as well as 42 major industrial sectors in China.

- Page 4, Line 84-87 (Introduction)

Domestically, the emission output of the Bitcoin mining industry would rank in the top 10 among 182 prefecture-level cities and 42 major industrial sectors in China, accounting for approximately 5.41% of the emissions of the electricity generation in China according to the China Emission Accounts & Datasets (www.ceads.net).

- Page 9, Line 188-189 (Results, Carbon emission flows of Bitcoin blockchain operation)

At the domestic level, the emission output of the Bitcoin mining industry would rank in the top 10 among 182 Chinese prefecture-level cities and 42 major industrial sectors.

Fig. 3 | Bitcoin industry energy consumption and carbon emission comparison. In Fig. 2, the energy consumption and carbon emission of Bitcoin industry are compared to national level emissions of other countries as well as to the emissions of domestic cities and industrial sectors in China. Annual energy consumption and ranking by countries (a) are obtained from cia.gov (www.cia.gov), carbon emission and ranking by countries (b) are collected from [global carbon atlas](http://globalcarbonatlas.org) (www.globalcarbonatlas.org). The carbon emission by Chinese cities (c) and industrial sectors (d) are obtained from China Emission Accounts and Datasets (www.ceads.net). Due to the unreleased or missing data in some database, the above energy consumption and carbon emission data are obtained for 2016 level.

- Page 10, Line 217-221 (Carbon emission flows of Bitcoin blockchain operation)

In the Benchmark scenario, the peak annualized emission output of the Bitcoin mining industry would make it the 10th largest emitting sector out of a total of 42 major Chinese industrial sectors. In particular, it would account for approximately 5.41% of the emissions of the electricity generation in China according to the China Emission Accounts & Datasets (www.ceads.net).

Comment 4: Figure 3d. A rapidly decreasing profit rate, should slow down bitcoin investments and this in turn should slow down the decrease in profit rate until it reaches a value close to zero. Instead what Figure 3d shows is that profits appear to reach a stable negative value i.e. losses, an unsustainable state for any investor for a long period of time. Do losses persist in the case when the model is simulate to 2050?

Answer: Thank you for pointing this out. We sincerely apologize for not providing more details in our figures, which may have caused a misunderstanding with our results. We have redrawn the Figure and updated our results in the revised manuscript. Figure 4e (Figure 3d in the original manuscript) actually presents the total accumulated profit for miners operating in China, not the profit rate of miners. Instead, the profit rate is the rate of change in the total accumulated profit. In our original manuscript, we did not make this clear and we sincerely apologize for the confusion it may have caused. As a result, we have added an additional figure (Figure 4h), which actually shows the actual profit rate (income) of miners. In Figure 4h, we can see that a rapidly decreasing profit rate does slow down bitcoin investments. In turn, it slows down the decrease in profit rate until it reaches a stable value close to zero. Although the total accumulated profit for the collective miners in China does end up being a negative value, the zero profit rate does indicate they are no longer losing money continuously and have reached a steady state. We hope this explanation is able to clarify the misunderstanding and confusion in the original manuscript, the specific content for this change is as follows:

- Page 11-12, Line 245-251 (Carbon policy effectiveness evaluation)

In the BM scenario, Bitcoin miners' profit rate are expected to drop to zero in April 2024, which suggests that the Bitcoin miners will gradually stop mining in China and relocate their

operations elsewhere. However, it is important to note that the entire relocation process does not occur immediately. Miners with higher sunk costs tends to stay in operation longer than those with lower sunk costs, hoping to eventually make a profit again. Consequently, the overall energy consumption associated with Bitcoin mining remains positive until the end of 2030, at which time almost all miners would have relocated elsewhere.

Fig. 4 | BBCE scenario assessment comparisons. a-i, monthly network energy consumption (a), carbon emission per GDP (b), carbon emission flows (c), network hash rate (d) miner cumulative profits (e) block hash difficulty (f), energy consumption cost (g), miner profit rate (h) and carbon emission cost (i) under each intended policy are simulated and calculated by BBCE framework. Based on the regressed parameters of the BBCE model, the whole sample timesteps of network carbon emission assessment cover the period from January 2014 to January 2030.

Comment 5: Figure 4. introduce plus and minus signs in line with standard system dynamics notation (Lane, 2000). Replace heavy energy with Coal based energy in all related variables. Replace Clean energy with hydro based energy. In general, strive to make variable names more specific and accurate as to what they represent in reality.

Lane DC. 2000. Diagramming conventions in system dynamics. Journal of the Operational Research Society 51(2), 241-245.

Answer: Thank you for pointing this out. We apologize for the confusions we have confused with our variable names. In the revised manuscript, we have strived to make them more specific and accurate to what they represent in reality. We have replaced heavy energy with coal-based energy, as well as replace clean energy with hydro-based energy for all instances in our revised manuscript. In addition, we have introduced plus and minus signs in line with standard system dynamics notations (Lane, 2000) in Figure 5 (in the revised manuscript). The specific changes are as follows:

▪ **Supplementary Table**

Supplementary Table 1 Variable descriptions				
Type	Parameter	Definition	Unit	Source
Level	Miner cumulative Profits	Total accumulated profits of Bitcoin miner in China	USD	-
	GDP	Gross productivity of Bitcoin blockchain in China	USD	-
Rate	Total Carbon Emission	Accumulated carbon emission of Bitcoin blockchain in China	kg	-
	Miner profit rate	Bitcoin miners' income per month	USD/month	-
	Investment intensity	Investment intensity of Bitcoin miners	-	Küfeoğlu & Özkuran ¹ ; CBECI
	GDP growth	Gross domestic product added per month	USD/month	-
Auxiliary	Carbon emission flow	Carbon emission of Bitcoin blockchain per month	Kg/month	-
	Mining hash rate	Mining hashes per second of Bitcoin network	Trillion hashes/second	BTC.com
	Mining efficiency	Average mining efficiency of Bitcoin network	Joule/ Trillion hashes	Küfeoğlu & Özkuran ¹ ; CBECI
	Mining power	Average mining power of Bitcoin network	Watt	-
	Network energy consumption	Monthly energy consumption of Bitcoin operations	Kilowatt hour	-

Market access standard for efficiency	Market access standards for Bitcoin miners' efficiency	100%	-
Power usage effectiveness	Energy usage effectiveness of Bitcoin mining centers	-	Stoll et al. ²
Coal-based energy consumption	Energy consumed by Bitcoin blockchain in Coal-based region	Kilowatt hour	-
Hydro-based energy consumption	Energy consumed by Bitcoin blockchain in hydro-rich region	Kilowatt hour	-
Coal-based energy carbon emission	Carbon dioxide generated by Coal-based region miners in Bitcoin blockchain	Kg	-
Hydro-based carbon emission	Carbon dioxide generated by Hydro-based region miners in Bitcoin blockchain	Kg	-
Carbon intensity of Coal-based energy	Emission factor of Coal-based energy in China	Kg/Kilowatt hour	Cheng et al. ³
Carbon intensity of Hydro-based energy	Emission factor of Hydro-based energy in China	Kg/Kilowatt hour	Cheng et al. ³
Miner site selection	proportions of Bitcoin server located in coal-based region	%	BTC.com
Carbon emission cost	Monthly carbon emission cost in Bitcoin blockchain	USD	-
Energy price	Average energy (electricity) price in China	USD/kwh	World Bank
Energy cost	Monthly energy (electricity) cost in Bitcoin blockchain	USD	-
Total mining operating cost	Sum of carbon cost and energy cost	USD	-
Carbon tax	Average taxation for industrial carbon emission	USD/Kg	World Bank
Block hash difficulty	Global block hash difficulty in Bitcoin blockchain	T	-
New block	New block generated by miners per month	-	-

Proportion of Chinese miners	The proportion of Chinese miners in global Bitcoin mining system	%	BTC.com; Küfeoğlu & Özkuran ¹
Block size	Bitcoin blockchain size per month	Megabyte	BTC.com
Transaction fee	Transaction fee per month	Bitcoin	BTC.com
Bitcoin Price	Periodical Bitcoin price	USD	-
Block reward	Monthly Bitcoin mined	Bitcoin	-
Mining Reward Halving mechanism	The mining reward Halving mechanism of Bitcoin	-	-

Comment 6: Furthermore use more meaningful variable names. For example, proportion of what? same for difficulty, efficiency, market access (standard?) and so on.

Answer: Thank you for this suggestion. We have carefully modified all the variable names to make them more meaningful and specific in the revised manuscript. For example, “Proportion” has been modified to “proportion of Chinese miners”. “Difficulty” has been modified to “Block hash difficulty” to represent the global hash difficulty in Bitcoin blockchain mining. “Efficiency” has been modified to “Mining efficiency” to represent the Average mining energy efficiency of Bitcoin network. “Market access” has been modified to “Market access stanfard for efficiency”. The comprehensive changes to our variable names are reflected throughout the manuscript and as follows:

▪ **Supplementary Table**

Supplementary Table 1 Variable descriptions				
Type	Parameter	Definition	Unit	Source
Level	Miner cumulative Profits	Total accumulated profits of Bitcoin miner in China	USD	-
	GDP	Gross productivity of Bitcoin blockchain in China	USD	-
	Total Carbon Emission	Accumulated carbon emission of Bitcoin blockchain in China	kg	-

Rate	Miner profit rate	Bitcoin miners' income per month	USD/month	-
	Investment intensity	Investment intensity of Bitcoin miners	-	Küfeoğlu & Özkuran ¹ ; CBECI
	GDP growth	Gross domestic product added per month	USD/month	-
	Carbon emission flow	Carbon emission of Bitcoin blockchain per month	Kg/month	-
Auxiliary	Mining hash rate	Mining hashes per second of Bitcoin network	Trillion hashes/second	BTC.com
	Mining efficiency	Average mining efficiency of Bitcoin network	Joule/ Trillion hashes	Küfeoğlu & Özkuran ¹ ; CBECI
	Mining power	Average mining power of Bitcoin network	Watt	-
	Network energy consumption	Monthly energy consumption of Bitcoin operations	Kilowatt hour	-
	Market access standard for efficiency	Market access standards for Bitcoin miners' efficiency	100%	-
	Power usage effectiveness	Energy usage effectiveness of Bitcoin mining centers	-	Stoll et al. ²
	Coal-based energy consumption	Energy consumed by Bitcoin blockchain in Coal-based region	Kilowatt hour	-
	Hydro-based energy consumption	Energy consumed by Bitcoin blockchain in hydro-rich region	Kilowatt hour	-
	Coal-based energy carbon emission	Carbon dioxide generated by Coal-based region miners in Bitcoin blockchain	Kg	-
	Hydro-based carbon emission	Carbon dioxide generated by Hydro-based region miners in Bitcoin blockchain	Kg	-
Carbon intensity of Coal-based energy	Emission factor of Coal-based energy in China	Kg/Kilowatt hour	Cheng et al. ³	

Carbon intensity of Hydro-based energy	Emission factor of Hydro-based energy in China	Kg/Kilowatt hour	Cheng et al. ³
Miner site selection	proportions of Bitcoin server located in coal-based region	%	BTC.com
Carbon emission cost	Monthly carbon emission cost in Bitcoin blockchain	USD	-
Energy price	Average energy (electricity) price in China	USD/kwh	World Bank
Energy cost	Monthly energy (electricity) cost in Bitcoin blockchain	USD	-
Total mining operating cost	Sum of carbon cost and energy cost	USD	-
Carbon tax	Average taxation for industrial carbon emission	USD/Kg	World Bank
Block hash difficulty	Global block hash difficulty in Bitcoin blockchain	T	-
New block	New block generated by miners per month	-	-
Proportion of Chinese miners	The proportion of Chinese miners in global Bitcoin mining system	%	BTC.com; Küfeoğlu & Özkuran ¹
Block size	Bitcoin blockchain size per month	Megabyte	BTC.com
Transaction fee	Transaction fee per month	Bitcoin	BTC.com
Bitcoin Price	Periodical Bitcoin price	USD	-
Block reward	Monthly Bitcoin mined	Bitcoin	-
Mining Reward Halving mechanism	The mining reward Halving mechanism of Bitcoin	-	-

Comment 7: State whether Figure 4 shows the complete structure of the model or a more aggregate/simplified version of it. This clarification should be made because it is not clear how the number of people involved in bitcoin investments grows due their attractiveness. In Figure 4, it appears that this is included somehow in variable Hash rate but there is no equation for it in the manuscript.

Answer: Thank you for pointing this out. We apologize for any confusion our lack of clarification may have caused. Figure 5 (Figure 4 in the original manuscript) does show the complete structure of the model, and we have added a clarification for this in the revised manuscript.

▪ Page 18, Line 397-399 (Methods), Supplementary Figures

By investigating the inner feedback loops and causalities of the systems, system dynamics model is able to capture the corresponding dynamic behaviors of system variables based on proposed scenarios^{33,34}. Supplementary Fig. 1 indicates the complete structure of BBCE modelling.

Supplementary Fig. 1 | Flow diagram of BBCE modelling. Parameters of the Bitcoin blockchain carbon emission system in Supplementary Fig. 1 are quantified in BBCE simulations, which are suggested by the feedback loops of Bitcoin blockchain. The whole quantitative relationships of BBCE parameters are demonstrated in Supplementary Methods.

Comment 8: Regarding equations 1-6: clarify what α , β are. Use meaningful variable names. e.g. proportion or efficiency is uninformative.

Answer: Thank you for pointing this out. We sincerely apologize for not clarifying the parameters in the equations of the the original manuscript. In the revised manuscript, we have added description and rationale for each parameter shown in the equations. For example, the parameter α_1 serves as the investment intensity function coefficient on the time and the proportion of Chinese miners; β_1 and α_2 represent the network hash rate constant function coefficient and coefficient on investment intensity, respectively; β_2 and α_3 indicate the block size function constant coefficient and coefficient on time, respectively; β_3 and α_4 act as the mining efficiency function constant coefficient and coefficient on investment intensity and market access standard for efficiency, respectively. In addition, we have adopted more meaningful variable names in the revised manuscript. For example, Proportion has been changed to Proportion of Chinese miners; Efficiency has been changed to Mining efficiency to represent the average mining efficiency of the Bitcoin network. The specific contents of this change are as follows:

- Page 20, Line 462-464 (Methods)

The initial value of static parameters in BBCE model are shown in Supplementary Table 2, the actual values of the parameterizations adopted are reported in Supplementary Methods, and the key quantitative settings of each subsystem are respectively run as follows:

- Page 20, Line 473-478 (Methods)

$$\text{Investment intensity} = \alpha_1 \times \text{Time} \times \text{Proportion of Chinese miners} \quad (1)$$

In Equation (1), the parameter α_1 serves as the investment intensity function coefficient on time and the proportion of Chinese miners, which is estimated and formulated by the historical data of Bitcoin blockchain operations from the period of January 2014 to January 2020.

- Page 20-21, Line 482-491 (Methods)

$$\text{Miner cumulative profits}_t = \int_0^t (\text{Miner profit rate} - \text{Investment intensity}) dt \quad (2)$$

As discussed above, the aim of Bitcoin mining hardware investment is to improve the miner's hash rate and the probability of broadcasting a new block. Utilizing the statistics of Bitcoin blockchain, the hash rate of the Bitcoin network is regressed, and the equation is:

$$\text{Mining hash rate} = e^{\beta_1 + \alpha_2 \text{Investment intensity}} \quad (3)$$

Where β_1 and α_2 represent the network hash rate constant function coefficient and coefficient on investment intensity, respectively.

- Page 21, Line 497-498 (Methods)

Where β_2 and α_3 indicate the block size function constant coefficient and coefficient on time, respectively.

- Page 21-22, Line 502-522 (Methods)

$$\text{Proportion of Chinese miners} = \text{IF THEN ELSE}(\text{Miner cumulative Profits} < 0, 0.7 - 0.01 \times \text{Time}, 0.7) \quad (5)$$

Suggested by the mining pool statistics obtained from BTC.com, China accounts for approximately 70% of Bitcoin blockchain operation around the world. As a result, we set the initial proportion of Chinese Bitcoin miners as 70%. In addition, the proportion of Chinese Bitcoin miners will gradually decrease if the Bitcoin mining process is no longer profitable in China.

The energy consumed per hash will reduce, i.e., the mining efficiency of the Bitcoin blockchain will improve when updated Bitcoin hardware is invested and introduced. Moreover, the market assess standard for efficiency proposed by policy makers also affects network efficiency. Consequently, the mining efficiency can be calculated as follows:

$$\text{Mining efficiency} = e^{\beta_3 + \alpha_4 \times \text{Investment intensity} \times \text{Market assess standard for efficiency}} \quad (6)$$

Where β_3 and α_4 act as the mining efficiency function constant coefficient and coefficient on investment intensity and market access standard for efficiency, respectively. The above function coefficients of BBCE parameters are regressed and formulated based on the actual Bitcoin blockchain operation data from the period of January 2014 to January 2020, and the specific value of each parameter is reported in Supplementary Methods.

Comment 9: All variable names should be consistent with Figure 4.

Answer: Thank you for pointing this out. We apologize for the inconsistencies of the variable names in the original manuscript. We have carefully modified all variable names so they are consistent with Figure 5 (Figure 4 in the original manuscript). The specific contents of the change are as follows:

- Page 17, Line 391-394 (Methods), Supplementary Figures

By investigating the inner feedback loops and causalities of the systems, system dynamics model is able to capture the corresponding dynamic behaviors of system variables based on proposed scenarios^{33,34}. Supplementary Fig. 1 indicates the complete structure of BBCE modelling.

Supplementary Fig. 1 | Flow diagram of BBCE modelling. Parameters of the Bitcoin blockchain carbon emission system in Supplementary Fig. 1 are quantified in BBCE simulations, which are suggested by the feedback loops of Bitcoin blockchain. The whole quantitative relationships of BBCE parameters are demonstrated in Supplementary Methods.

Comment 10: What does 0.7 represent in equation (5)? Why is this parameter value chosen?

Answer: Thank you for pointing this out. We apologize for not clarifying the specific meaning of this coefficient in equation (5). 0.7 represents the initial proportion of Chinese Bitcoin miners. As suggested by the mining pool statistics obtained from BTC.com, China accounts for approximately 70% of Bitcoin blockchain operation around the world. Consequently, we set the initial proportion of Chinese Bitcoin miners to 0.7. We have added an explanation to clarify the meaning of this parameter value in the revised manuscript. The specific content of this change is as follows:

- Page 21, Line 502-509 (Methods)

$$\text{Proportion of Chinese miners} = \text{IF THEN ELSE}(\text{Miner cumulative Profits} < 0, 0.7 - 0.01 \times \text{Time}, 0.7) \quad (5)$$

Suggested by the mining pool statistics obtained from BTC.com, China accounts for approximately 70% of Bitcoin blockchain operation around the world. As a result, we set the initial proportion of Chinese Bitcoin miners as 70%. In addition, the proportion of Chinese Bitcoin miners will gradually decrease if the Bitcoin mining process is no longer profitable in China.

Comment 11: Provide the rationale for equation 11, what do the values 0.01, 1, 2 represent?

Answer: Thank you for pointing this out. We apologize for not clarifying the rationale for equation (11). As suggested by the World Bank database, we introduce the average taxation percentage for industrial carbon emission (1%) as the initial carbon tax parameter in BBCE modelling. Furthermore, if the carbon emission per GDP of the Bitcoin blockchain mining operations is larger than the average industrial carbon emission per GDP in China, which is approximately 2kg/GDP (the first 2 in the equation), the policy maker will conduct punitive carbon taxation actions by doubling the carbon taxation on the Bitcoin blockchain (the second 2 in the equation). Otherwise, the carbon taxation on the Bitcoin blockchain would remain the same as other industries (the 1 in the equation). Therefore, 0.01 represents the average taxation percentage for industrial carbon emission as the initial carbon tax parameter in the BBCE modeling. The first 2 represents the approximate average industrial carbon emission per GDP in China. The second 2 in the equation represents the carbon taxation doubling action by the policy maker. 1 represents the normal carbon taxation action. We have added an explanation for equation (11) in the revised manuscript. The specific content of this change is as follows:

- Page 23, Line 546-553 (Methods)

Suggested by the World Bank database, we introduce the average taxation percentage for industrial carbon emission (1%) as the initial carbon tax parameter in BBCE modelling. In addition, the punitive carbon taxation on the Bitcoin blockchain will be conducted by policy makers, i.e, the carbon taxation on the Bitcoin blockchain will be doubled, if the carbon emission per GDP of the

Bitcoin blockchain is larger than average industrial carbon emission per GDP in China (2 kg/GDP). As a result, the carbon tax of Bitcoin blockchain is set as:

$$\text{Carbon tax} = 0.01 \times \text{IF THEN ELSE} (\text{carbon emission per GDP} > 2, 2, 1) \quad (11)$$

Comment 12: The model has been constructed in vensim software.

Answer: Thank you for pointing this out. Our model has been constructed in vensimPLE software, and we have added a sentence to clarify this in the revised manuscript. The specific content of this change is as follows:

- Page 20, Line 454-455 (Methods)
Our BBCE model has been constructed in Vensim software (PLE version)

Comment 13: To improve model transparency, it is standard practice to submit the model documentation using the SDM tool <https://www.systemdynamics.org/SDM-doc>

Answer: Thank you for this suggestion. We have included the model documentation using the SDM tool in the revised manuscript. The SDM model assessment provides assessment results in three categories: model information, warnings, and potential omissions. The information allows modelers and model readers to gain a better and specific understanding of the suitability of the model in terms of its elements and confidence building tests. The specific content of this change is as follows:

- **Supplementary Discussion**

Structural suitability tests. In order to improve model transparency and conduct structural suitability tests on BBCE modelling, the System Dynamics Model Documentation and Assessment Tool (SDM) is introduced to provide documentation of models built using the Vensim modeling software. The SDM model assessment provides assessment results in three categories: model information, warnings, and potential omissions. The above information allows modelers and

model readers to gain a better and specific understanding of the suitability of model in terms of its elements and confidence building tests³⁹.

Model Information	Result
Total Number Of Variables	38
Total Number Of State Variables	3 (7.9%)
Total Number Of Stocks	3 (7.9%)
Total Number Of Feedback Loops No IVV (Maximum Length: 30) [3..15]	17 (0 0 17)
Total Number Of Feedback Loops With IVV (Maximum Length: 30) [0..0]	0 (0 0 0)
Total Number Of Causal Links	51 (0 0 51)
Total Number of Rate-to-rate Links	1
Number Of Units Used In The Model (Basic/Combined)	2/0
Total Number Of Equations Using Macros	0 (0.0%)
Variables With Source Information	0 (0.0%)
Dimensionless Unit Variables	0 (0.0%)
Function Sensitivity Parameters	0 (0.0%)
Data Lookup Tables	0 (0.0%)
Time Unit	Month
Initial Time	1
Final Time	204
Reported Time Interval	TIME STEP
Time Step	1
Model Is Fully Formulated	Yes

Warnings	Result
Variables Not In Any View	0 (0.0%)
Nonmonotonic Lookup Functions	0 (0.0%)
Cascading Lookup Functions	0 (0.0%)
Non-Zero End Sloped Lookup Functions	0 (0.0%)
Equations With If Then Else Functions	2 (5.3%)
Equations With Min Or Max Functions	0 (0.0%)
Equations With Step Pulse Or Related Functions	2 (5.3%)

Potential Omissions	Result
Unused Variables	0 (0.0%)
Supplementary Variables	0 (0.0%)
Supplementary Variables Being Used	0 (0.0%)
Complex Variable	1 (2.6%)
Complex Stock	0 (0.0%)

Supplementary Fig. 2 | Model assessment results of BBCE modelling. Based on the System Dynamics Model Documentation and Assessment Tool, Fig 6 presents the basic BBCE modelling assessment results. The whole assessment results are demonstrated in Supplementary Materials.

Supplementary Fig. 2 provides the basic BBCE modelling assessment results based on SDM tool. The structural suitability test results indicate that proposed BBCE model is able to effectively reflect the causal relationship and feedback loops in Bitcoin carbon emission system: all of the key variables are covered, and the causal relationship between variables is appropriate; the model boundary is comparatively appropriate; all the system parameters of the BBCE model have practical significance.

Comment 14: In the revised version submit the output of the SDM tool as supplementary material.

Answer: Thank you for this suggestion. We have submitted the output of the SDM tool as supplementary material in the revised version of our manuscript. Please see Supplementary Materials for the complete output.

Comment 15: Line 570: you refer to the results of model validation. Provide the graphs, on hash rate and efficiency.

Answer: Thank you for pointing this out. We apologize for not providing the results of model validation in the original manuscript. To assess the difference between real historical behaviors and BBCE modelling simulations, the reality and statistical test are performed for the actual mining hash rate and mining efficiency. We introduce R^2 to interpret the goodness of fit and parameter consistencies of BBCE modelling. We introduce We have provided the graphs on hash rate and efficiency in the revised manuscript. The specific changes are as follows:

- **Supplementary Discussion**

Reality and statistical tests. To assess the difference between real historical behaviors and BBCE modelling simulations, the reality and statistical test are performed by comparing the projected data with historical time-series data. The key Bitcoin blockchain operating time-series data from the period of January 2014 to January 2020, including actual mining hash rate and mining efficiency, are utilized to verify the parameter consistencies of BBCE modelling. We introduce R^2 to interpret the goodness of fit and parameter consistencies of BBCE modelling. Suggested by the pervious studies^{40,41}, the reality and statistical results are generally considered to be acceptable if the R^2 is greater than 0.9.

Supplementary Fig. 3 | Reality and statistical test results. Fig 8 illustrates the historical and projected mining hash rate (a) and mining efficiency (b) comparison results based on the actual bitcoin time-series data. We introduce R^2 to interpret the goodness of fit and parameter consistencies of BBCE modelling.

As shown in Supplementary Fig. 3, the estimated mining hash rate and mining efficiency are compared to their historical time-series data. The results show that the R^2 of estimated mining hash rate and mining efficiency are all greater than 0.9, at 0.97 and 0.91 respectively. The reality and statistical testing results indicate that the proposed BBCE model has a superior consistency between model behavior and actual situation, and also illustrate the behavioral realities of the BBCE parameters.

Text Remarks

Comment 16: Line 35: takes

Answer: Thank you for pointing this out. We apologize for the grammatical error in the original manuscript. We have carefully corrected the grammatical errors and typos in the revised version of our manuscript. The specific content of the change is as follows:

- Page 2, Line 34-35 (Introduction)

In this paper, we quantify the current and future carbon emission patterns of Bitcoin blockchain operations in China under different carbon policies.

Comment 16: Line 42: capture and reproduce the endogenous dynamics of complex system elements (Sterman, 2000; Richardson, 2011).

Answer: Thank you for pointing this out. We apologize for the grammatical error in the original manuscript. We have carefully modified the sentence in the revised version of our manuscript. The specific content of the change is as follows:

- Page 2, Line 39-41 (Introduction)

system dynamics technique is able to capture and reproduce the endogenous dynamics of complex system elements, which enables the simulation and estimation of specific industry operations^{6,7,8}.

Comment 17: Line 62: From the text it is evident that PUE should stand for power usage efficiency.

Answer: Thank you for pointing this out. We have made this correction in the revised manuscript. The specific content of the change is as follows:

- Page 3, Line 59-60 (Introduction)

second, power usage efficiency (PUE) is introduced to illustrate the energy consumption efficiency of Bitcoin blockchain as suggested by Stoll¹³.

Comment 18: Line 96: assesses

Answer: Thank you for pointing this out. We apologize for this grammatical error and have made the correction in the revised manuscript. The specific change is as follows:

- Page 4, Line 89 (Introduction)

Through scenario analysis, we find that

Comment 19: Line 128: replace closeness with proximity

Answer: Thank you for this suggestion. We have replaced closeness with proximity in the revised manuscript. The specific change is as follows:

- Page 5, Line 123-124 (Results, The energy and carbon emission problem of PoW algorithm in China)

Due to the proximity to manufacturers of specialized hardware and access to cheap electricity,

Comment 20: Line 132: replace with: China is a key signatory of the Paris agreement

Answer: Thank you for this suggestion. We have made this replacement in the revised manuscript. The specific change is as follows:

- Page 5, Line 127 (Results, The energy and carbon emission problem of PoW algorithm in China)

China is a key signatory of the Paris Agreement.

Comment 21: Line 144: start sentence with: As suggested...

Answer: Thank you for this suggestion. We have made this replacement in the revised manuscript. The specific change is as follows:

- Page 6, Line 139-140 (Results. The energy and carbon emission problem of PoW algorithm in China)

As suggested by the actual regional statistics of Bitcoin miners,

Comment 22: Line 149: market access standard for efficiency

Answer: Thank you for this suggestion. We have made the replacement in the revised manuscript. The specific change is as follows:

- Page 7, Line 144-145 (Results, The energy and carbon emission problem of PoW algorithm in China)
market access standard for efficiency is doubled,

Comment 23: Line 151: what does matian mean? Replace with another word.

Answer: Thank you for pointing this out. We apologize for the typo in the original manuscript. We have replaced the word with “maintain” in the revised manuscript.

- Page 7, Line 146-147 (Results, The energy and carbon emission problem of PoW algorithm in China)
policy makers are forced to maintain the network stability of Bitcoin blockchain in an efficient manner.

Comment 24: Line 169: replace maximize with peak

Answer: Thank you for this suggestion. We have replaced maximize with peak in the revised manuscript.

- Page 7, Line 164-165 (Results, Carbon emission flows of Bitcoin blockchain operation)
In the BM scenario, the annual energy consumption of Bitcoin blockchain in China will gradually grow and eventually peak in 2024, at 296.59 Twh per year.

Comment 25: Figure 2: change the format to that of figure 3 for consistency

Answer: Thank you for pointing this out. We have changed the format of Figure 2 to be consistent with that of Figure 3 in the revised manuscript. The specific change is as follows:

- Page 8, Line 173-180 (Results, Carbon emission flows of Bitcoin blockchain operation)

Fig. 2 | Annualized scenario simulation results. Annualized energy consumption (a) and carbon emission flows (b) of Bitcoin operation in China are generated through monthly simulation results of BBCE modelling. The blue, red, yellow and green bars in (a) and (b) indicate the annual energy consumption and carbon emission flows of Chinese Bitcoin industry in benchmark, site regulation, market access and carbon tax scenarios separately. Each plot is presented as point estimates (solid bar) and 95% confidence intervals (error bars).

Comment 26: Line 249: replace attracting with attractive

Answer: Thank you for this suggestion. We have replaced attracting with attractive in the revised manuscript.

- Page 12, Line 265 (Results, Carbon policy effectiveness evaluation)

Some attractive conclusions can be drawn based on the results of BBCE simulation

Comment 27: Line 251: this is an obscure sentence. What does emissions prompted policy mean? Replace emission reduced with emission reduction.

Answer: Thank you for pointing this out. We sincerely apologize for this confusion in the original manuscript. We want to express that the MA scenario actually raises, rather than reduces, the emission output based on the simulation outcome. In the revised manuscript, we have rewritten this sentence more clearly in the revised manuscript. The specific change is as follows:

- Page 12, Line 265-267 (Results, Carbon policy effectiveness evaluation)
although the MA scenario enhances the market access standard to increase Bitcoin miners' efficiencies, it actually raises, rather than reduces, the emission output based on the simulation outcome.

Comment 28: Line 256: the surviving miners

Answer: Thank you for this suggestion. We have made this replacement in the revised manuscript.

- Page 12, Line 271-272 (Results, Carbon policy effectiveness evaluation)
However, the surviving miners are all devoted to squeezing more proportion of the network hash rate

Comment 29: Line 258: the bitcoin industry in China generates more...

Answer: Thank you for this suggestion. We have made this replacement in the revised manuscript.

- Page 12, Line 273 (Results, Carbon policy effectiveness evaluation)
In addition, the bitcoin industry in China generates more CO₂ emissions under the MA scenario.

3 Responses to Reviewer #3:

We sincerely appreciate your valuable feedback on our manuscript. All of these comments have helped us significantly in improving the quality of our manuscript. Based on your comments and suggestions, we have made extensive changes to the manuscript in the revised manuscript. Our responses to your specific comments are provided below. We also provide a copy of the relevant sections (marked in red) following our responses.

Major Comments

Comment 1: (iv) Model clarity: The verbal description of the SD modelling of Bitcoin Blockchain Carbon Emissions (BBCE) does not allow the reader to understand the contribution relative to the existing literature: e.g. (a) GDP does not measure productivity, but measures based on ‘changes in GDP’ like GDP growth, does.

Answer: Thank you for pointing this out. We apologize for the confusing verbal descriptions of the SD modelling of Bitcoin Blockchain Carbon Emissions (BBCE) in the original manuscript. We have carefully modified the variable names and their respective descriptions to provide clarification for the reader. For example, “GDP” has been changed to “GDP growth” to measure the Gross domestic product added per month. The comprehensive changes are shown as follows:

- **Supplementary Table**

Supplementary Table 1 Variable descriptions				
Type	Parameter	Definition	Unit	Source
Level	Miner cumulative Profits	Total accumulated profits of Bitcoin miner in China	USD	-
	GDP	Gross productivity of Bitcoin blockchain in China	USD	-
	Total Carbon Emission	Accumulated carbon emission of Bitcoin blockchain in China	kg	-
Rate	Miner profit rate	Bitcoin miners’ income per month	USD/month	-

	Investment intensity	Investment intensity of Bitcoin miners	-	Küfeoğlu & Özkuran ¹ ; CBECI
	GDP growth	Gross domestic product added per month	USD/month	-
	Carbon emission flow	Carbon emission of Bitcoin blockchain per month	Kg/month	-
Auxiliary	Mining hash rate	Mining hashes per second of Bitcoin network	Trillion hashes/second	BTC.com
	Mining efficiency	Average mining efficiency of Bitcoin network	Joule/ Trillion hashes	Küfeoğlu & Özkuran ¹ ; CBECI
	Mining power	Average mining power of Bitcoin network	Watt	-
	Network energy consumption	Monthly energy consumption of Bitcoin operations	Kilowatt hour	-
	Market access standard for efficiency	Market access standards for Bitcoin miners' efficiency	100%	-
	Power usage effectiveness	Energy usage effectiveness of Bitcoin mining centers	-	Stoll et al. ²
	Coal-based energy consumption	Energy consumed by Bitcoin blockchain in Coal-based region	Kilowatt hour	-
	Hydro-based energy consumption	Energy consumed by Bitcoin blockchain in hydro-rich region	Kilowatt hour	-
	Coal-based energy carbon emission	Carbon dioxide generated by Coal-based region miners in Bitcoin blockchain	Kg	-
	Hydro-based carbon emission	Carbon dioxide generated by Hydro-based region miners in Bitcoin blockchain	Kg	-
	Carbon intensity of Coal-based energy	Emission factor of Coal-based energy in China	Kg/Kilowatt hour	Cheng et al. ³
	Carbon intensity of Hydro-based energy	Emission factor of Hydro-based energy in China	Kg/Kilowatt hour	Cheng et al. ³

Miner site selection	proportions of Bitcoin server located in coal-based region	%	BTC.com
Carbon emission cost	Monthly carbon emission cost in Bitcoin blockchain	USD	-
Energy price	Average energy (electricity) price in China	USD/kwh	World Bank
Energy cost	Monthly energy (electricity) cost in Bitcoin blockchain	USD	-
Total mining operating cost	Sum of carbon cost and energy cost	USD	-
Carbon tax	Average taxation for industrial carbon emission	USD/Kg	World Bank
Block hash difficulty	Global block hash difficulty in Bitcoin blockchain	T	-
New block	New block generated by miners per month	-	-
Proportion of Chinese miners	The proportion of Chinese miners in global Bitcoin mining system	%	BTC.com; Küfeoğlu & Özkuran ¹
Block size	Bitcoin blockchain size per month	Megabyte	BTC.com
Transaction fee	Transaction fee per month	Bitcoin	BTC.com
Bitcoin Price	Periodical Bitcoin price	USD	-
Block reward	Monthly Bitcoin mined	Bitcoin	-
Mining Reward Halving mechanism	The mining reward Halving mechanism of Bitcoin	-	-

Comment 2: (b) The allowed feedback loops presumably give incentives to Bitcoin miners (through reduced incomes, approximated by GDP) to reduce investments in mining (e.g. upgrading antMiners) and/or mining efforts, everything else constant. However, that would imply a reduced power consumption, not an increased one (to reach the projected 296.59 TW/h by 2024, and generate 130.50 MtCO₂e), which is why the contribution is unclear (further Fig. 5 could not be found, and instead the reader assumed that Fig. 4 –Methods section-- provided a flow diagram of the SD model structure: please correct me if wrong. Appendix B provides the list of BBCE model equations).

Answer: Thank you for pointing this out. First, we sincerely apologize for the mistake in Figure labeling in the original manuscript. The flow diagram of the SD model structure is now correctly labeled as Fig. 5 in the revised manuscript. The allowed feedback loops do provide incentives to Bitcoin miners to reduce investments in mining, however, as shown in Equation (12), this reduction in total investment intensity is mainly attributed to a reduced proportion of Chinese miners in operation. In Equation (13) in Supplementary Methods, we shown that the proportion of Chinese miners gradually decreases as the Miner Cumulative Profit drops below zero. In other words, as long as the cumulative profit is positive, miners have no incentives to leave the mining operation in China. While they are in operation, any reduction in individual investment intensity would put miners in disadvantage, which jeopardizes their chances of mining new blocks and receiving the reward (Tschorsch & Scheuermann, 2016). Therefore, we make the assumption that miners maintain full individual investment intensity (e.g. upgrading antMiners) as long as they are mining new blocks. Consequently, as long as there is no reduction in the proportion of Chinese miners, the total investment intensity would not decrease. In our simulation results shown in Fig. 4(e), we can see that the miner cumulative profit approaches the zero around July of 2024. As a result, the power consumption increases and peaks around this period. After this period, the cumulative profit does turn negative and the proportion of Chinese miners begins to decrease, which in turn reduces the total investment intensity and the network energy consumption. Again, we apologize for the confusion and we hope this explanation provides clarification to our results. In the revised manuscript, we have clearly stated this assumption, the specific content of this change is as follows:

- Page 17, Line 387-389 (Methods)

(5) Miners maintain full investment intensity while in operation, as any reduction in individual investment intensity would put miners in disadvantage and jeopardize their chances of mining new blocks and receiving the reward.

Tschorsch, F., & Scheuermann, B. Bitcoin and beyond: A technical survey on decentralized digital currencies. *IEEE Commun. Surv. Tutor.* 18, 2084-2123 (2016).

Comment 3: (c) The actual values of the parameterizations adopted are unreported, e.g. in equations (1), (3), (4), (6), etc. the values of the parameters $\alpha_1, 2, 3, 4$ and/or $\beta_1, 2, 3$ are never reported, reducing the reader's ability to comment on reasonableness and/or replicability.

Answer: Thank you for pointing this out. We sincerely apologize for not reporting the actual values of the parameterization adopted in the original manuscript. These specific values are reported in Supplementary Methods, and we have added clarification in the revised manuscript to clearly inform readers about this point. To further enhance the reader's ability to comment on reasonableness and replicability, we have added description and rationale for each parameter shown in the equations. For example, the parameter α_1 serves as the investment intensity function coefficient on the time and the proportion of Chinese miners; β_1 and α_2 represent the network hash rate constant function coefficient and coefficient on investment intensity, respectively; β_2 and α_3 indicate the block size function constant coefficient and coefficient on time, respectively; β_3 and α_4 act as the mining efficiency function constant coefficient and coefficient on investment intensity and market access standard for efficiency, respectively. The specific contents of this change are as follows:

- Page 20, Line 462-464 (Methods)

The initial value of static parameters in BBCE model are shown in Supplementary Table 2, the actual values of the parameterizations adopted are reported in Supplementary Methods, and the key quantitative settings of each subsystem are respectively run as follows:

- Page 21, Line 474-478 (Methods)

$$\text{Investment intensity} = \alpha_1 \times \text{Time} \times \text{Proportion of Chinese miners} \quad (1)$$

In Equation (1), the parameter α_1 serves as the investment intensity function coefficient on time and the proportion of Chinese miners, which is estimated and formulated by the historical data of Bitcoin blockchain operations from the period of January 2014 to January 2020.

- Page 20-21, Line 482-491 (Methods)

$$\text{Miner cumulative profits}_t = \int_0^t (\text{Miner income} - \text{Investment intensity}) dt \quad (2)$$

As discussed above, the aim of Bitcoin mining hardware investment is to improve the miner's hash rate and the probability of broadcasting a new block. Utilizing the statistics of Bitcoin blockchain, the hash rate of the Bitcoin network is regressed, and the equation is:

$$\text{Mining hash rate} = e^{\beta_1 + \alpha_2 \text{Investment intensity}} \quad (3)$$

Where β_1 and α_2 represent the network hash rate constant function coefficient and coefficient on investment intensity, respectively.

- Page 21, Line 497-498 (Methods)

Where β_2 and α_3 indicate the block size function constant coefficient and coefficient on time, respectively.

- Page 21-22, Line 502-528 (Methods)

$$\text{Proportion of Chinese miners} = \text{IF THEN ELSE}(\text{Miner cumulative Profits} < 0, 0.7 - 0.01 \times \text{Time}, 0.7) \quad (5)$$

Suggested by the mining pool statistics obtained from BTC.com, China accounts for approximately 70% of Bitcoin blockchain operation around the world. As a result, we set the initial proportion of Chinese Bitcoin miners as 70%. In addition, the proportion of Chinese Bitcoin miners will gradually decrease if the Bitcoin mining process is no longer profitable in China.

The energy consumed per hash will reduce, i.e., the mining efficiency of the Bitcoin blockchain will improve when updated Bitcoin hardware is invested and introduced. Moreover, the market assess standard for efficiency proposed by policy makers also affects network efficiency. Consequently, the mining efficiency can be calculated as follows:

$$\text{Mining efficiency} = e^{\beta_3 + \alpha_4 \times \text{Investment intensity} \times \text{Market assess standard for efficiency}} \quad (6)$$

Where β_3 and α_4 act as the mining efficiency function constant coefficient and coefficient on investment intensity and market access standard for efficiency, respectively. The above function

coefficients of BBCE parameters are regressed and formulated based on the actual Bitcoin blockchain operation data from the period of January 2014 to January 2020, and the specific value of each parameter is reported in Supplementray Methods.

The mining power of the Bitcoin blockchain can be obtained by network hash rate and mining efficiency. The equation of mining power is shown as follows:

$$\text{Mining power} = \text{Mining hash rate} \times \text{Mining efficiency} \quad (7)$$

- Page 22, Line 534-539 (Methods)

Employed the regional data of Bitcoin mining pools, coal-based and hydro-based energy is proportionally consumed by distinctive Bitcoin pools. The total carbon flows in Bitcoin blockchain are measured by the sum of both monthly coal-based and hydro-based energy carbon emission growth. The integration of total carbon emission is:

$$\text{Total carbon emission}_t = \int_0^t \text{Carbon emission flow } dt \quad (9)$$

- Page 23, Line 546-553 (Methods)

Suggested by the World Bank database, we introduce the average taxation percentage for industrial carbon emission (1%) as the initial carbon tax parameter in BBCE modelling. In addition, the punitive carbon taxation on the Bitcoin blockchain will be conducted by policy makers, i.e, the carbon taxation on the Bitcoin blockchain will be doubled, if the carbon emission per GDP of the Bitcoin blockchain is larger than average industrial carbon emission per GDP in China (2 kg/GDP). As a result, the carbon tax of Bitcoin blockchain is set as:

$$\text{Carbon tax} = 0.01 \times \text{IF THEN ELSE} (\text{carbon emission per GDP} > 2, 2, 1) \quad (11)$$

Comment 4: (d) Some recent research reports that it is important to understand seasonal movements of miners within China to take advantage of cheap surplus energy availability due to the rain season (e.g. Stoll et al., 2019). According to Fig. 4, this would correspond to introducing a causal arrow from either the electricity price or the electricity cost to miners' location decisions,

which is currently absent. If I wanted to have a sense of the quantitative impact of this more realistic scenario on the reported authors' point predictions for emissions, it remains unclear from the current presentation efforts how to think about it? And it would certainly change the estimates obtained under scenario SR, in Table 1.

Answer: Thank you for pointing this out. In the revised manuscript, we have taken the seasonal movements of miners within China to take advantage of cheap surplus energy availability into account by introducing a casual arrow from the energy (electricity) price to miners' location decisions. While this change did change the estimates obtained under scenario SR, SR scenario still yielded the lowest carbon emission out of all the scenarios. The specific changes of this content is as follows:

- Page 7, Line 147-150 (The energy and carbon emission problem of PoW algorithm in China)

In the Site Regulation (SR) scenario, Bitcoin miners in the coal-heavy area are persuaded and suggested to relocate to the hydro-rich area to take advantage of the relatively lower cost of surplus energy availability in the area due to factors such as rain season, which results in only 20% of miners remaining in coal-heavy areas in the scenario.

- Page 17, Line 397-399 (Methods), Supplementary Figures

By investigating the inner feedback loops and causalities of the systems, system dynamics model is able to capture the corresponding dynamic behaviors of system variables based on proposed scenarios^{33,34}. Supplementary Fig. 1 indicates the complete structure of BBCE modelling.

Supplementary Fig. 1 | Flow diagram of BBCE modelling. Parameters of the Bitcoin blockchain carbon emission system in Supplementary Fig. 1 are quantified in BBCE simulations, which are suggested by the feedback loops of Bitcoin blockchain. The whole quantitative relationships of BBCE parameters are demonstrated in Supplementary Methods.

Comment 5: (v) Model validation is so synthetic that I could not understand, e.g. what does ‘0.9, at 0.977 and 0.913 respectively,’ (in lines 570-1 of the manuscript) mean? The authors should provide at least the definition and critical values of the statistical tests used to come up with statements such as this, because model validation is crucial for rendering the policy evaluation exercises credible. Another way to validate the novel model predictions would be to circumscribe it to the time window for which recent research reports estimates of Bitcoin power consumption and associated carbon emissions (e.g. Stoll et al., 2019, report in a figure their estimates in relation to those obtained until 2018, at an annual frequency; CBECI provides an online tool against which the authors’ model based results could be benchmarked, etc.). More concretely: In Appendix C it is stated that ‘It is estimated that between the period of January 1st, 2016 and June 30th, 2018, up to 13 million metric tons of CO₂ emissions can be attributed to the Bitcoin Blockchain’ which are

certainly below the estimates reported in recent publications (e.g. Stoll et al., 2019 report about 22 MtCO₂ for 2018 alone). Yet, the SD-modelling departs from such low point estimates, and forecasts them to increase almost 25-fold by 2024?

Answer: Thank you for pointing this out. We sincerely apologize for the synthetic model validation in the original manuscript. In order to assess the difference between real historical behaviors and BBCE modelling simulations, we conduct a more extensive result and statistical test by comparing the projected data with historical time-series data. The key Bitcoin blockchain operating time-series data, including actual mining hash rate and mining efficiency, are utilized to verify the parameter consistencies of BBCE modelling. We use R^2 to interpret the goodness of fit and parameter consistencies of BBCE modelling. The results indicate that the R^2 of estimated mining hash rate and mining efficiency are all greater than 0.9, at 0.97 and 0.91 respectively. The comprehensive results of the statistical test are shown in Part 2 of Supplementary Discussion. The specific contents of the change are as follows:

- **Supplementary Discussion**

Reality and statistical tests. To assess the difference between real historical behaviors and BBCE modelling simulations, the reality and statistical test are performed by comparing the projected data with historical time-series data. The key Bitcoin blockchain operating time-series data from the period of January 2014 to January 2020, including actual mining hash rate and mining efficiency, are utilized to verify the parameter consistencies of BBCE modelling. We introduce R^2 to interpret the goodness of fit and parameter consistencies of BBCE modelling. Suggested by the pervious studies^{40,41}, the reality and statistical results are generally considered to be acceptable if the R^2 is greater than 0.9.

Supplementary Fig. 3 | Reality and statistical test results. Fig 8 illustrates the historical and projected mining hash rate (a) and mining efficiency (b) comparison results based on the actual bitcoin time-series data. We introduce R^2 to interpret the goodness of fit and parameter consistencies of BBCE modelling.

As shown in Supplementary Fig. 3, the estimated mining hash rate and mining efficiency are compared to their historical time-series data. The results show that the R^2 of estimated mining hash rate and mining efficiency are all greater than 0.9, at 0.97 and 0.91 respectively. The reality and statistical testing results indicate that the proposed BBCE model has a superior consistency between model behavior and actual situation, and also illustrate the behavioral realities of the BBCE parameters.

Comment 6: (vi) Robustness of the model to alternative parameterizations, e.g. a power usage of electricity (PUE) of 1.10 is adopted citing Stoll et al. (2019), without further discussion. But one wonders how robust the reported estimates (and results on the ranking of the different policies considered) would be to more realistic measures of 1.05 or 1.0 (e.g. for pools in Inner Mongolia, where cooling is unnecessary). Similarly, one of the main sources of uncertainty in estimating carbon emissions, are the actual carbon intensities of different sources of electricity, and the reader

wonders how robust the reported point estimates are to alternative carbon intensities (or alternative proportions of miners using ‘clean’ energy sources, in terms of scenario SR).

Answer: Thank you for pointing this out. We certainly agree robustness of the model is vital for providing information about the reliability of the model. In the revised manuscript, we have conducted detailed sensitivity analysis to examine the robustness of our BBCE model. For example, we have set the power usage of efficiency (PUE) to 1.15 and 1.05 with respect to the utilized PUE of 1.1 in the BBCE model. Furthermore, we have set the proportion of miners using ‘clean’ energy sources to 43% (23% in Site Regulation scenario) and 37% (17% in Site Regulation scenario) with respect to the original proportion of 40% (20% in Site Regulation scenario). The sensitivity test on BBCE modelling shows that alternative key parameter values do not lead to remarkable changes in the model behaviors or ranking of the intended carbon reduction policies, which indicates that the proposed BBCE model has excellent behavioral robustness and stability. The specific content of this change is as follows:

- **Supplementary Discussion**

Sensitivity analysis. Sensitivity analysis examines the robustness of BBCE model. By Adjusting the settings of important parameters, we can comment on the robustness and stability of BBCE modelling in terms of long-term trend of carbon emission flows and the carbon emission ranking of the different policies. Two key constant parameters of BBCE model, i.e., power usage of efficiency (PUE) and proportions of Chinese Bitcoin servers located in coal-based region (Miner site selection) are introduced to conduct sensitivity analysis. Specifically speaking, we set PUE at 1.15 and 1.05 with respect to the utilized PUE of 1.1 in BBCE model, and Miner site selection at 43% (23% in Site Regulation scenario) and 37% (17% in Site Regulation scenario) regarding to original Miner site selection at 40% (20% in Site Regulation scenario).

Supplementary Fig. 4| Sensitivity analysis results. (a)-(d) provide alternative initial parameter settings of Power Usage of Efficiency (PUE) in each simulation scenario from the whole sample period and comparisons of the estimated carbon emission flows under different parameterizations. The red dash lines in (a)-(d) denote parameterization of PUE at 1.15 and the green dash lines at 1.05. (e)-(h) provides alternatives initial proportions of Chinese Bitcoin servers located in coal-based region in each scenario. The red dash lines in (e)-(h) denote parameterization of proportions of Chinese Bitcoin servers located in coal-based region at 43% (at 23% in Site Regulation scenario) and the green dash lines at 37% (at 17% in Site Regulation scenario). The blue solid lines from (a)-(d) denote the parameterizations of PUE at 1.1 in each scenario, and that of (e)-(h) denote the parameterizations of proportions of Chinese Bitcoin servers located in coal-based region at 40% in each scenario (20% in Site regulation scenario, which are utilized in the actual BBCE modelling).

Supplementary Fig. 4 reports the sensitivity and robustness results of carbon emission flow in each scenario. It is clear that the carbon emission flow is directly proportional to the power usage of electricity (PUE) and proportions of Chinese Bitcoin servers located in coal-based region (Miner site selection). However, the long-term carbon emission trends of each sensitivity settings are consistent with that of the original BBCE parameterizations. In addition, Site Regulation scenario stable generates the lowest carbon emission flows among the 4 scenarios under different parameterizations, which indicates its stable carbon emission reduction effectiveness on the Chinese Bitcoin industry. Overall, the sensitivity test on BBCE modelling shows that alternative key parameter values do not lead to remarkable changes in the model behaviors or ranking of the intended carbon reduction policies, thus indicating that the proposed BBCE model has excellent behavioral robustness and stability.

Comment 7: (vii) Reliability of reported point estimates: policy decision makers typically worry about the percentage chances that a given policy is going to have unintended consequences, i.e. about how confident the researcher is when recommending some policy change on the basis of

her/his model. This is typically encapsulated in some notion of ‘reliability’ of the reported mean forecast or point estimate, as measured by their prediction intervals (PI) (e.g. with a 95% chance, emissions are not going to be higher than the upper bound of the PI, nor lower than the lower bound). The authors should provide some ‘reliable bounds’ associated with their point estimates (and results of their alternative policy scenarios), or explicitly warn the reader about the difficulty of doing so, providing a valid reason.

Answer: Thank you for pointing this out. We have incorporated reliable bounds (95%) confidence intervals on our point estimates and results of our alternative policy scenarios in our revised manuscript. The specific content of this change is as follows:

- Page 8, Line 174-180 (Results, Carbon emission flows of Bitcoin blockchain operation)

Fig. 2 | Annualized scenario simulation results. Annualized energy consumption (a) and carbon emission flows (b) of Bitcoin operation in China are generated through monthly simulation results of BBCE modelling. The blue, red, yellow and green bars in (a) and (b) indicate the annual energy consumption and

carbon emission flows of Chinese Bitcoin industry in benchmark, site regulation, market access and carbon tax scenarios separately. Each plot is presented as point estimates (solid bar) and 95% confidence intervals (error bars).

Comment 8: In relation to the above points, some stated claims need further clarification, like (line 234): ‘In the BM scenario, Bitcoin miner profits are expected to drop to zero in April 2024, which suggests that the Bitcoin miners will gradually stop mining in China and relocate their operations elsewhere’. If that is the case, it is unclear how the overall energy consumption associated with Bitcoin mining can carry on being positive between April 2024 and the end of 2030, as reported in the top-right panel of Figure 3, i.e. better understanding of the model, baseline model parameters and estimated regression parameters, is crucial to gauge whether the reported predictions make actual sense, lending further credibility to the policy experiments considered.

Answer: Thank you for pointing this out. We sincerely apologize for not providing more details in our figures, which may have caused a misunderstanding with our results. Figure 3e (Figure 3d in the original manuscript) actually presents the total accumulated profit for miners operating in China, not the profit rate of miners. As the total accumulated profit for the miners turns negative around 2024, the proportion of miners operating in China begins to decrease. However, not all miners leave at once - miners with relatively lower sunk cost would be the first ones to leave the operation. Meanwhile, miners with relatively higher sunk cost tend to stay for longer as they hope to recover the losses. Therefore, there are still miners remaining between 2024 and 2030, which means that the overall energy consumption associated with Bitcoin mining remains positive. In our revised manuscript, we have added an additional figure (Figure 3h), which actually shows the actual profit rate (income) of miners. In Figure 3h, we can see that between 2024 and 2030, the profit rate first experiences a decrease and then an increase until it reaches a stable value close to zero. When the profit rate has reached a stable equilibrium of zero, almost all miners would have left China and relocated elsewhere, which implies that the overall energy consumption associated with Bitcoin mining is no longer positive. We hope this explanation is able to clarify the misunderstanding and confusion in the original manuscript, the specific content for this change is as follows:

- Page 11, Line 245-264 (Results, Carbon policy effectiveness evaluation)

In the BM scenario, Bitcoin miners' profit rate are expected to drop to zero in April 2024, which suggests that the Bitcoin miners will gradually stop mining in China and relocate their operations elsewhere. However, it is important to note that the entire relocation process does not occur immediately. Miners with higher sunk costs tends to stay in operation longer than those with lower sunk costs, hoping to eventually make a profit again. Consequently, the overall energy consumption associated with Bitcoin mining remains positive until the end of 2030, at which time almost all miners would have relocated elsewhere.

Fig. 4 | BBCE scenario assessment comparisons. a-i, monthly network energy consumption (a), carbon emission per GDP (b), carbon emission flows (c), network hash rate (d) miner accumulative profits (e) block hash difficulty (f), energy consumption cost (g), miner profit rate (h) and carbon emission cost (i) under each intended policy are simulated and calculated by BBCE framework. Based on the regressed parameters of the BBCE model, the whole sample timesteps of network carbon emission assessment cover the period from January 2014 to January 2030.

Minor Comments

Comment 9: Needs serious English proof-reading.

Answer: Thank you for pointing this out. We sincerely apologize for the English mistakes in the original manuscript. We have carefully proof-read and corrected the grammatical errors in the revised manuscript.

Comment 10: Some references cited do not support the written sentence, e.g. (line 133) ‘However, without appropriate interventions and feasible policies, the intensive Bitcoin blockchain operations in China can quickly grow as a threat that could potentially undermine the emission reduction effort taken place in the country¹⁰’. But then reference No. 10 refers to Ebola? Redding, D. W., Atkinson, P. M., Cunningham, A. A., Iacono, G. L., Moses, L. M., Wood, J. L., & Jones, K. E. Impacts of environmental and socio-economic factors on emergence and epidemic potential of Ebola in Africa. *Nat. Commun.* 10, 1-11 (2019).

Answer: Thank you for pointing this out. We sincerely apologize for this mistake in the original manuscript. We have carefully checked to make sure all references cited are supportive of the written sentence. For example, reference 10 has been replaced with the following:

▪ Page 26, Line 596-597 (References)

10. Wu, S., Liu, L., Gao, J., & Wang, W. Integrate risk from climate change in China under global warming of 1.5 and 2.0° C. *Earth's Future*, 7(12), 1307-1322 (2019).

REVIEWERS' COMMENTS

Reviewer #1 (Remarks to the Author):

I am pleased with how you incorporated my comments and I have no further remarks on your manuscript.

Reviewer #2 (Remarks to the Author):

I would like to thank the authors for the effort put in manuscript revision.

I have one comment concerning the use of the SDM tool.

The output of the tool indicates whether the model has equations with unit errors. This is a standard test for model validity.

The warning section of the tool output in the pdf you provided does not include this.

You should provide evidence that the model has no unit errors for it to be accepted.

Please provide the complete sdm output in its original html format. See the warning section in the sample output below.

<http://wayback.archive->

it.org/10432/20181121203235/http://lm.systemdynamics.org/tools/sdm/Handbook%20Model-A.html

Reviewer #3 (Remarks to the Author):

I would like to thank the authors for carefully taking into account my (and other) referees' comments in their revised submitted manuscript. From my perspective, the following minor sticking points remain:

-Although the readability of the resubmitted draft has improved, I would still advise further proof-reading, e.g. lines 38-39, replace [.] in '[system dynamics technique] is' by [SD]; line 58, replace [.] in 'miners are [invested]' by [mining]; lines 61-62, delete [.] in 'the dynamic[s] behaviour[s] of Bitcoin miner[']s [investment]'; etc.

-Neither in the abstract nor in the discussion sections of the current version the main conclusions of the policy assessments conducted are clearly formulated. Yet I regard the latter as the main contribution of the paper, i.e. that moving away from the current carbon tax policy consensus to a 'site regulation' (SR) policy may reduce the overall carbon emissions associated with Bitcoin mining. The authors should be explicit about this, as well as to the limitations of the assumptions leading to this important conclusion, e.g. that these are simulations arising from SD modelling, and that the SR scenario assumes no cost on miners from relocating to 'clean-energy' based regions.

Signed: Dr. H.F. Calvo-Pardo

Detailed Response to Reviewers

We would like to express our heartfelt gratitude to three anonymous reviewers for their valuable feedback. Our manuscript, titled “Policy assessments for the carbon emission flows and sustainability of Bitcoin blockchain operation in China”, benefited significantly from the constructive comments and insights from the review team. Based on the suggestions received, we have made careful revisions to the original manuscript. In the revised manuscript, all changes are marked in red. In addition, we have also carefully proofread this manuscript for typographical, grammatical, and other errors. We hope the revised manuscript is able to meet your standard of quality and address the concerns raised by the reviewers. In the following, you will find our detailed point-by-point responses to the reviewers’ comments.

1 Responses to Reviewer #1

We sincerely appreciate all your valuable feedback on our manuscript during the revision process. All of the comments have helped us significantly in improving the quality of our manuscript. Thank you.

2 Responses to Reviewer #2

We are grateful for your valuable feedbacks on our manuscript during the revision process. All of the comments have helped us significantly in improving the quality of our manuscript. Based on your comment and suggestion, we have made changes to the manuscript in the revised manuscript. Our response to your specific comment is provided below.

Minor Comments

Comment 1: I would like to thank the authors for the effort put in manuscript revision. I have one comment concerning the use of the SDM tool. The output of the tool indicates whether the model has equations with unit errors. This is a standard test for model validity. The warning section of the tool output in the pdf you provided does not include this. You should provide evidence that the model has no unit errors for it to be accepted. Please provide the complete sdm output in its original html format. See the warning section in the sample output below. <http://wayback.archive-it.org/10432/20181121203235/http://lm.systemdynamics.org/tools/sdm/Handbook%20Model-A.html>

Answer: Thank you for pointing this out. We have carefully checked to make sure the model has no unit errors. We also provide evidence and the complete SDM output in its original html format in the Supplementary Discussion (see Supplementary Fig.3) and Supplementary Notes.

- **Supplementary Fig. 3**

Model Assessment Results

Model Information	Result
Total Number Of Variables	38
Total Number Of State Variables	3 (7.9%)
Total Number Of Stocks	3 (7.9%)
Total Number Of Feedback Loops No IVV (Maximum Length: 30) [3, 15]	17 (0 0 17)
Total Number Of Feedback Loops With IVV (Maximum Length: 30) [0, 0]	0 (0 0 0)
Total Number Of Causal Links	51 (0 0 51)
Total Number Of Rate-to-rate Links	1
Number Of Units Used In The Model (Basic/Combined)	3/0
Total Number Of Equations Using Macros	0 (0.0%)
Variables With Source Information	0 (0.0%)
Dimensionless Unit Variables	17 (44.7%)
Function Sensitivity Parameters	0 (0.0%)
Data Lookup Tables	0 (0.0%)
Time Unit	Month
Initial Time	1
Final Time	204
Reported Time Interval	TIME STEP
Time Step	1
Model Is Fully Formulated	Yes

Warnings	Result
Variables Not In Any View	0 (0.0%)
Nonmonotonic Lookup Functions	0 (0.0%)
Cascading Lookup Functions	0 (0.0%)
Non-Zero End Sloped Lookup Functions	0 (0.0%)
Equations With If Then Else Functions	2 (5.3%)
Equations With Min Or Max Functions	0 (0.0%)
Equations With Step Pulse Or Related Functions	2 (5.3%)
Equations With Unit Errors Or Warnings	0 (0.0%)

Potential Omissions	Result
Unused Variables	0 (0.0%)
Supplementary Variables	0 (0.0%)
Supplementary Variables Being Used	0 (0.0%)
Complex Variable	1 (2.6%)
Complex Stock	0 (0.0%)

Supplementary Fig. 3 | Model assessment results of BBCE modelling. Based on the System Dynamics Model Documentation and Assessment Tool, this Figure presents the basic BBCE modelling assessment results. The whole assessment results are demonstrated in Supplementary Notes.

3 Responses to Reviewer #3:

We sincerely appreciate your valuable feedback on our manuscript during the revision process. All of the comments have helped us significantly in improving the quality of our manuscript. Based on your comments and suggestions, we have made careful changes in the revised manuscript. Our responses to your specific comments are provided below. We also provide a copy of the relevant sections (marked in red) following our responses.

Minor Comments

Comment 1: Although the readability of the resubmitted draft has improved, I would still advise further proof-reading, e.g. lines 38-39, replace [.] in '[system dynamics technique] is' by [SD]; line 58, replace [.] in 'miners are [invested]' by [mining]; lines 61-62, delete [.] in 'the dynamic[s] behaviour[s] of Bitcoin miner[']s [investment]'; etc.

Answer: Thank you for pointing this out. We have carefully proof-read and corrected the grammatical errors in the revised manuscript. For example:

- Page 2, Line 42-43 (Introduction)
SD is able to capture and reproduce the endogenous dynamics of complex system elements,
- Page 3, Line 62 (Introduction)
when high hash rate miners are mining;
- Page 3, Line 65-66 (Introduction)
which further influences the dynamic behavior of Bitcoin miners.

Comment 2: Neither in the abstract nor in the discussion sections of the current version the main conclusions of the policy assessments conducted are clearly formulated. Yet I regard the latter as the main contribution of the paper, i.e. that moving away from the current carbon tax policy consensus to a 'site regulation' (SR) policy may reduce the overall carbon emissions

associated with Bitcoin mining. The authors should be explicit about this, as well as to the limitations of the assumptions leading to this important conclusion, e.g. that these are simulations arising from SD modelling, and that the SR scenario assumes no cost on miners from relocating to ‘clean-energy’ based regions.

Answer: Thank you for pointing this out. We have explicitly stated the main results of the policy assessments in the Abstract and Discussion sections of the revised manuscript. In addition, we have also noted the possible limitations of assumptions for our results in the discussion section. The specific changes are as follows:

- Page 1, Line 15-25 (Abstract)

In this work, we show that moving away from the current punitive carbon tax policy to a “site regulation” policy which induces changes in the energy consumption structure of the mining activities is more effective in limiting carbon emission of Bitcoin blockchain operation.

- Page 15, Line 342-346 (Discussion)

Through scenario analysis, we show that moving away from the current punitive carbon tax policy consensus to a site regulation (SR) policy which induces changes in the energy consumption structure of the mining activities is more effective in limiting the total amount of carbon emission of Bitcoin blockchain operations.

- Page 15, Line 357-363 (Discussion)

Furthermore, our site regulation (SR) scenario assumes no cost on miners from relocating to “clean-energy” based regions. In reality, there may be certain costs associated with this action, such as transportation. Therefore, although our results suggest that a site regulation (SR) policy may be more effective than the current punitive carbon tax policy consensus in limiting the total amount of carbon emission of Bitcoin blockchain operations, it is important to note that these are simulations arising from system dynamics modeling and are limited by these assumptions.